# Disco🌐: Densely-overlapping Cell Instance Segmentation via Adjacency-aware Collaborative Coloring

**Rui Sun**[1,2,3]*, **Yiwen Yang**[1,4,5]*, **Kaiyu Guo**[1], **Chen Jiang**[1,2]†, **Dongli Xu**[1], **Zhaonan Liu**[6], **Tan Pan**[1,2], **Limei Han**[1,2], **Xue Jiang**[2], **Wu Wei**[5]†, **Yuan Cheng**[1,2,3]†

[1]Shanghai Academy of Artificial Intelligence for Science, [2]Fudan University
[3]Shanghai Innovation Institute, [4]School of Life Sciences and Biotechnology, Shanghai Jiao Tong University
[5]Lingang Laboratory, [6]Renji Hospital, School of Medicine, Shanghai Jiao Tong University

## Abstract

Accurate cell instance segmentation is foundational for digital pathology analysis. Existing methods based on contour detection and distance mapping still face significant challenges in processing complex and dense cellular regions. Graph coloring-based methods provide a new paradigm for this task, yet the effectiveness of this paradigm in real-world scenarios with dense overlaps and complex topologies has not been verified. Addressing this issue, we release a large-scale dataset GBC-FS 2025, which contains highly complex and dense sub-cellular nuclear arrangements. We conduct the first systematic analysis of the chromatic properties of cell adjacency graphs across four diverse datasets and reveal an important discovery: most real-world cell graphs are non-bipartite, with a high prevalence of odd-length cycles (predominantly triangles). This makes simple 2-coloring theory insufficient for handling complex tissues, while higher-chromaticity models would cause representational redundancy and optimization difficulties. Building on this observation of complex real-world contexts, we propose **Disco** (**D**ensely-overlapping Cell **I**nstance **S**egmentation via Adjacency-aware **CO**llaborative Coloring), an adjacency-aware framework based on the "divide and conquer" principle. It uniquely combines a data-driven topological labeling strategy with a constrained deep learning system to resolve complex adjacency conflicts. First, "Explicit Marking" strategy transforms the topological challenge into a learnable classification task by recursively decomposing the cell graph and isolating a "conflict set." Second, "Implicit Disambiguation" mechanism resolves ambiguities in conflict regions by enforcing feature dissimilarity between different instances, enabling the model to learn separable feature representations. Disco achieves a significant 7.08% improvement in the PQ metric on the GBC-FS 2025 dataset and an average improvement of 2.72% across all datasets. Furthermore, the predicted "Conflict Map" serves as a novel tool for interpreting topological complexity, offering new potential for data-driven pathology research. The code is publicly available at https://github.com/SR0920/Disco.

## 1 Introduction

Accurate cell instance segmentation has long been a foundational task for cell counting, morphological analysis, spatial histology studies, and even cancer grading Zhang et al. (2025a) Zhang et al. (2025b). However, due to the intrinsic complexity of biological tissues, cells often exhibit highly dense, severely overlapping, and morphologically diverse characteristics, making the precise disambiguation of each individual instance while maintaining high accuracy a formidable challenge. Although mainstream methods based on detection Jiang et al. (2023), contour prediction Chen et al.

---

*These authors contributed equally
†Corresponding authors: jiangchen@sais.org.cn, wuwei@lglab.ac.cn, cheng_yuan@fudan.edu.cn

**Figure 1.** Visual comparison of mainstream instance segmentation paradigms with our proposed Disco framework. (a) Detection-based methods are constrained by coarse bounding box representations and heuristic non-maxima suppression (NMS). (b) Contour-based methods are highly sensitive to binarization thresholds. (c) Distance-based methods rely on complex post-processing for instance reconstruction. (d) Disco framework reformulates the problem by directly modeling the cell adjacency graph, ultimately reconstructing instances through topological decoding.

(2016), and distance/direction maps He et al. (2021) have made significant progress in specific scenarios, they share a common reliance on local pixelwise or geometric information to infer instance affiliation. As illustrated in Figure 1 (a-c), this strategy reveals its inherent fragility when processing complex cell clusters. The common bottleneck of these approaches is their lack of an intrinsic mechanism to explicitly model the global topological constraints among cells, thus their decisions are inherently locally optimal and prone to systematic errors in complex cell clusters.

To overcome these limitations, methods based on graph coloring theory Fritsch et al. (1998) have recently emerged, offering a new paradigm with global topological awareness. Among these, 2-coloring, based on bipartite graph theory Asratian et al. (1998), represents the most concise and efficient model. Concurrently, the pioneering work of FCIS Zhang et al. (2025c) demonstrated the potential of a universal 4-coloring model. However, the validity of its core bipartite assumption on real-world cellular topologies has remained a critical and unverified gap. To rigorously investigate this issue, we make two foundational contributions. First, we introduce **GBC-FS 2025** (**G**all**b**ladder **C**ancer **F**rozen **S**ection **2025** Dataset), a new large-scale stress-test dataset comprising 2,839 frozen section images of gallbladder cancer with 864,204 annotated sub-cellular nuclei, which exhibits highly complex and dense sub cellular structures. Second, utilizing this dataset and three other public benchmarks, we conduct the first systematic analysis of the chromatic properties of cell adjacency graphs. As shown in Table 1, our analysis reveals a striking discovery: real-world cell graphs are fundamentally non-bipartite, characterized by a high prevalence of odd-length cycles (predominantly 3-cycles). This finding suggests that any simple model relying on the ideal bipartite graph assumption may face fundamental limitations when processing complex histopathology images. These findings call for a fundamentally new approach that can elegantly handle both bipartite and non-bipartite structures.

Table 1: A Cross-Dataset Comparison of Key Topological Properties.

| Dataset | Proportion of Cells with $\leq$ 3 Neighbors | Proportion of 3-Cycles among Odd Cycles | Bipartite Node Ratio | Conflict Node Ratio | Secondary Conflict Node Ratio |
|---|---|---|---|---|---|
| PanNuke | 100.00% | 0.00% | 100.00% | 0.00% | 0.00% |
| DSB2018 | 99.66% | 97.94% | 98.01% | 1.99% | 1.84% |
| CryoNuSeg | 99.11% | 98.12% | 94.36% | 5.64% | 4.82% |
| GBC-FS 2025 | 88.09% | 90.51% | 69.51% | 30.49% | 24.64% |

Facing this severe and underestimated challenge, we propose **Disco** (**D**ensely-overlapping Cell **I**nstance **S**egmentation via Adjacency-aware **CO**llaborative Coloring), a conflict-aware and dynamic 2-coloring framework. Disco is a new method that fundamentally rethinks cell instance segmentation through advanced graph-theoretic coloring. We do not resort to a more general high-chromaticity model, as this would be an unnecessary redundancy for the simple bipartite structures that still dominate the graph. Instead, we propose a more refined "divide and conquer" strategy. The core of the Disco framework is comprised of two innovative mechanisms: "Explicit Marking," which transforms topological conflicts into a learnable classification target, and "Implicit Disambiguation," which resolves the intrinsic ambiguities of discrete labels in the continuous feature space. In summary, the contributions of this paper are four-fold:

- *Groundbreaking Topological Analysis*: We conduct the first systematic, quantitative analysis of the chromatic properties of real-world cell adjacency graphs, disruptively revealing their fundamentally non-bipartite nature characterized by dense "conflict clusters." This finding provides a critical, data-driven theoretical foundation for applying graph coloring paradigms to cell segmentation.

- *A Novel Disco Framework*: Based on these findings, we propose Disco, a conflict-aware framework operating on a "divide and conquer" principle. It uniquely synergizes two core mechanisms: "Explicit Marking," which transforms topological conflicts into a structured, learnable target, and "Implicit Disambiguation," which resolves discrete label ambiguities in the continuous feature space via an end-to-end adjacency constraint.

- *A High-density Case Study Dataset*: We introduce GBC-FS 2025, which contains over 850,000 annotated instances of sub-cellular nuclei data. With its unprecedented cell density and extreme topological complexity, including a conflict node ratio exceeding 30%, it provides an indispensable "stress-test" platform for the development of robust segmentation algorithms.

- *SOTA Performance and a New Interpretability Paradigm*: Disco achieves state-of-the-art (SOTA) performance across four heterogeneous datasets, with a remarkable 7.08% PQ improvement on GBC-FS 2025. Furthermore, we pioneer the use of the predicted "Conflict Map" as a novel topological quantification tool, opening new avenues for data-driven pathology research.

## 2 FROM LOCAL CUES TO GLOBAL TOPOLOGY

The evolution of cell instance segmentation methods can be understood as a paradigm shift from approaches reliant on local geometric information to those that embrace global topological structures Chen et al. (2024) Petukhov et al. (2022). To provide a comprehensive understanding of existing methods, we conduct a summary analysis herein. At the same time, the Appendix A.2 provides a more extensive review of related works.

### 2.1 BOTTLENECKS OF LOCAL CUES

Mainstream instance segmentation paradigms Wang et al. (2025), while diverse in their technical implementations, are fundamentally united by their reliance on local, pixel-wise or geometric cues to infer instance affiliation. Detection-based methods, epitomized by Mask R-CNN He et al. (2017), operate on coarse geometric bounding boxes. Their final performance is often dictated by the heuristic nature of non-maxima suppression (NMS) Ren et al. (2016), which frequently leads to missed instances in crowded scenes with irregular morphologies or high degrees of overlap (Figure 1(a)). Contour-based methods Xu et al. (2024) are highly sensitive to the binarization threshold, often resulting in instance merging (under-segmentation) at low thresholds and fragmentation (over-segmentation) at high thresholds (Figure 1(b)). Distance/direction-based methods, such as StarDist Schmidt et al. (2018) and Hover-Net Graham et al. (2019), attempt to mitigate these issues by learning richer representations, but they in turn depend on complex and error-prone post-processing algorithms to reconstruct instances from the predicted vector fields, making them susceptible to error propagation that can cause erroneous instance splitting (Figure 1(c)). The common bottleneck of these methods is their lack of an intrinsic mechanism to comprehend the global topological structure among cells. Their decisions are, by design, locally optimal, which inevitably leads to systematic failures when faced with the global complexity of dense tissues.

### 2.2 PROMISE OF GLOBAL TOPOLOGY

To overcome this fundamental limitation, a new perspective that abstracts the instance segmentation problem into a graph coloring task has emerged Anh et al. (2020), thereby explicitly modeling these global topological constraints. The most concise instantiation within this paradigm is 2-coloring Zhao et al. (2025), based on the elegant theory of bipartite graphs. This model is theoretically sufficient for any graph that contains no odd-length cycles. This raises a critical, yet unexamined, question: to what extent do real-world cell graphs adhere to this ideal bipartite structure? Concurrently, the pioneering work of FCIS Zhang et al. (2025c) demonstrated the potential of a universal 4-coloring model based on the Four-Color Theorem. However, this approach risks introducing unnecessary representational redundancy and potential optimization difficulties when the underlying

**Figure 2.** Fundamental topological structures in cell adjacency graphs. (a) Simple, 2-colorable bipartite structures. (b) Non-bipartite structures containing odd-length cycles (e.g., 3-cycles), which induce coloring conflicts. (c) A complex "conflict cluster" formed by interconnected odd cycles, leading to secondary conflicts between adjacent conflict nodes.

topology is simpler. This presents a "Goldilocks challenge" Hulubei et al. (2003): that of finding a coloring model that is neither too simple nor too complex. These considerations necessitate a systematic topological analysis to uncover the true nature of cellular graphs, which in turn motivates our quest for an optimally balanced, dynamically adaptive approach.

## 3 CROSS-DATASET TOPOLOGICAL ANALYSIS

A principled solution necessitates a profound understanding of the problem's intrinsic structure. While graph-based paradigms offer a promising new perspective for instance segmentation, the topological properties of the resulting cell adjacency graphs have remained a largely unexplored domain Liu et al. (2022). To fill this critical gap, we conduct a systematic, quantitative analysis across four datasets of significant heterogeneity to empirically characterize the topological landscape of real-world cellular tissues. This analysis not only provides a solid foundation for our proposed Disco framework but also constitutes one of our core scientific contributions.

### 3.1 GRAPH FORMULATION AND TOPOLOGICAL PRELIMINARIES

We formally model the spatial arrangement of cells as a Cell Adjacency Graph (CAG) Trémeau & Colantoni (2000).

**Definition 1. Cell Adjacency Graph, CAG**: Given an instance mask $I \in \mathbb{Z}^{H \times W}$ comprising a set of instances $\mathcal{S} = \{s_1, ..., s_N\}$, its corresponding CAG is an undirected graph $G = (V, E)$, where the vertex set $V = v_1, ..., v_N$ is in bijective correspondence with $\mathcal{S}$ and the edge set is defined as $E = (v_i, v_j) \mid s_i, s_j \in \mathcal{S}, i \neq j,$ and $\mathcal{N}(s_i) \cap s_j \neq \emptyset$. Here, $\mathcal{N}(s_i)$ denotes the pixel set of instance $s_i$ after a morphological dilation with a $3 \times 3$ kernel, a definition that captures all 8-connected adjacencies.

The feasibility of resolving instance disambiguation via graph coloring is fundamentally tied to the graph's chromatic number $\chi(G)$, for which bipartiteness is a key concept.

**Theorem 1. Bipartite Graph Theorem Csóka et al. (2016)**: A graph $G$ is 2-colorable (i.e., $\chi(G) \leq 2$) if and only if it is bipartite, which is equivalent to stating that $G$ contains no odd-length cycles.

This theorem establishes a direct link between a graph's structure—the presence of odd cycles—and the minimum number of colors required to resolve all adjacency constraints. As illustrated in Figure 2, cellular arrangements can form both simple, 2-colorable bipartite structures and more complex, non-bipartite structures containing odd cycles. The presence of a single odd cycle necessitates at least three colors, thereby creating a coloring conflict for any 2-coloring scheme. In dense tissues, multiple odd cycles may share vertices, forming conflict clusters. In such cases, multiple conflict-inducing nodes can themselves be adjacent, a situation we term secondary conflicts. We now proceed to quantitatively investigate the prevalence of these structures in real-world data.

### 3.2 EMPIRICAL ANALYSIS OF GRAPH PROPERTIES

We analyzed the CAGs derived from four datasets with varying complexities, and the comprehensive results summarized in Figure 3. Our analysis focuses on two key aspects: local connectivity and global topology.

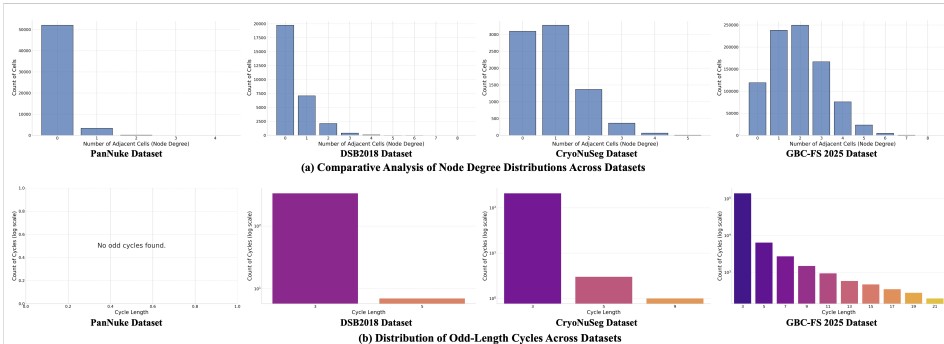

**Figure 3.** A Cross-dataset comparative analysis of cell adjacency graph topologies. (a) Node Degree Distributions: Local connectivity varies significantly, from the highly sparse PanNuke to the densely clustered GBC-FS 2025. (b) Odd-Length Cycle Distributions: The prevalence of 3-cycles confirms that most real-world cell graphs are non-bipartite, with complexity peaking in the GBC-FS 2025 dataset. The y-axis is on a logarithmic scale.

**Local Connectivity**: The node degree, i.e., the number of its neighbors Chen & Su (2023), reflects the local crowdedness of a cell. Figure 3(a) illustrates the node degree distributions for each dataset. Although the low average node degrees suggest overall graph sparsity, the long-tailed distributions and the presence of nodes with a maximum degree of up to 8 indicate the existence of highly dense local neighborhoods. As detailed in Table 1, in simpler datasets like DSB2018 and CryoNuSeg, over 99% of cells have three or fewer neighbors. In our more complex in-house GBC-FS 2025 dataset, however, this proportion drops to 88.09%, confirming a significant increase in local connectivity.

**Global Topology and Non-Bipartiteness**: Our most striking finding stems from the analysis of odd cycles. We found that, with the exception of the extremely sparse and entirely bipartite PanNuke dataset, non-bipartite graphs are a prevalent feature. Non-bipartite graphs constitute 56.67% of the images in CryoNuSeg and 29.17% in GBC-FS 2025. Figure 3(b) displays the length distribution of these odd cycles, revealing a strikingly consistent pattern: across all non-bipartite datasets, 3-cycles (triangles) overwhelmingly account for over 90% of all odd cycles. This provides strong, direct evidence for the argument that local structures formed by the direct contact of three or more cells are the primary drivers of topological complexity Vazquez et al. (2004).

## 3.3 THE PERVASIVENESS OF COLORING CONFLICTS

The high prevalence of odd cycles Nikiforov (2008) directly implies that simple 2-coloring models are infeasible. To quantify this, we introduce the concept of conflict nodes.

**Definition 2. Conflict Nodes and Conflict Set**: For a graph $G$, its conflict set $V_{conf}$ is a minimum vertex set such that the subgraph induced by its removal $G[V \backslash V_{conf}]$, is bipartite. The nodes in $V_{conf}$ are termed conflict nodes. The Conflict Node Ratio is given by $|V_{conf}|/|V|$.

As shown in our feasibility analysis in Table 1, this ratio varies dramatically across datasets, from 0% in PanNuke to a staggering 30.49% in GBC-FS 2025. This indicates that in complex tissues, nearly a third of all cells are involved in topological conflicts and cannot be resolved by a 2-coloring scheme. Furthermore, our analysis reveals a deeper challenge—secondary conflicts.

**Definition 3. Secondary Conflict**: In a graph $G = (V, E)$, given its conflict set $V_{conf}$, a secondary conflict is an edge $e = (u, v) \in E$ such that $u \in V_{conf}$ and $v \in V_{conf}$.

Our secondary conflict analysis reveals that these conflict nodes are not isolated. Secondary conflicts exist in all datasets that contain non-bipartite graphs. In the challenging GBC-FS 2025 dataset, the secondary conflict node ratio reaches 24.64%.

These quantitative results lead to an unequivocal conclusion: real-world cell adjacency graphs, particularly those from dense histopathology images, are fundamentally non-bipartite and are characterized by dense clusters of interconnected odd cycles. Further details regarding the dataset's analytical results and additional visualizations are provided in the Appendix A.3.

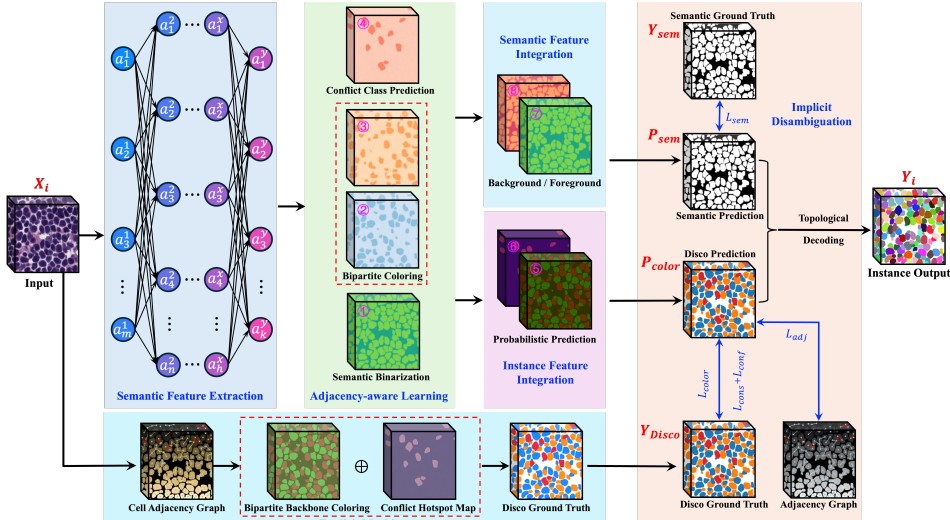

**Figure 4.** An overview of the training framework for our proposed Disco method. This framework synergistically integrates (1) data-driven topological analysis, which generates a Disco ground truth map $Y_{Disco}$, encoding both bipartite structures and conflict hotspots; (2) a dual-branch segmentation network, which learns to predict a foundational semantic map $P_{sem}$, and a detailed Disco coloring map $P_{color}$; and (3) a decoupled loss system, which provides targeted supervision. Crucially, the Adjacency Constraint Loss ($\mathcal{L}_{adj}$) leverages the ground truth adjacency graph to enforce feature dissimilarity between all neighboring instances in the continuous probability space, thereby enabling end-to-end constrained optimization.

## 4 METHOD

Our topological analysis has revealed a profound characteristic of real-world Cell Adjacency Graphs (CAGs): they are simultaneously dominated by simple, local bipartite arrangements yet punctuated by critical, non-bipartite conflict clusters. To better address this complex and variable reality, we propose Disco, a novel framework engineered to be both efficient for simple topologies and robust for complex ones. It eschews monolithic, high-chromaticity solutions in favor of a more elegant "divide and conquer" philosophy, which is actualized through two cornerstone mechanisms: "Explicit Marking" for label generation and "Implicit Disambiguation" for model optimization. Our framework implements a complete end-to-end learning process, synergistically integrating topological analysis, predictive modeling, and constrained learning, as illustrated in Figure 4.

### 4.1 THE "DIVIDE AND CONQUER" PRINCIPLE

The core of the Disco method is a "divide and conquer" strategy applied to the graph coloring problem. Instead of treating the CAG as a homogeneous entity, we first stratify the problem space based on its intrinsic topological structure.

Theoretically, the graph coloring problem is equivalent to partitioning the vertex set $V$ of a graph $G$ into a minimum number of disjoint Independent Sets $\{V_1, V_2, ..., V_k\}$. While finding this minimum partition, defined by the chromatic number $\chi(G)$, is an NP-Hard problem, our empirical analysis of the datasets provides a key insight: CAGs are predominantly dominated by a massive bipartite subgraph Alon (1996).

This crucial structural property is the foundation of our "divide and conquer" principle. It suggests that a monolithic, high-chromaticity model is suboptimal, as it applies the same complex machinery to both the simple, vast bipartite regions and the sparse, complex non-bipartite regions. Instead, we propose that an optimal strategy should first efficiently handle the dominant bipartite component and then apply a specialized mechanism to the isolated, topologically challenging conflict nodes. This principle allows us to design a more targeted and efficient learning framework, which we detail in the following sections.

**Figure 5.** A visual decomposition of the Disco label generation process.

## 4.2 EXPLICIT MARKING: CONFLICT-AWARE LABEL GENERATION

The "Explicit Marking" strategy transforms the raw instance ground truth into a topologically-informed supervisory signal, a process visually decomposed in Figure 5. This strategy operationalizes our "divide and conquer" principle through a highly efficient, two-stage decomposition process.

Our algorithm employs a Breadth-First Search (BFS) to efficiently extract the maximal bipartite subgraph identified in our analysis. This partitions the vast majority of "simple" nodes into two large independent sets, $V_1$ and $V_2$, which correspond to our two primary colors.

Theoretically, this process could continue recursively on the remaining subgraph, $G_{rem} = G[V \setminus (V_1 \cup V_2)]$. However, we make a critical, pragmatic design choice based on the Principle of Sufficient Representation. Our analysis shows that the remaining nodes, which we consolidate into a single conflict set $V_{conf}$, are few in number but form dense, topologically complex "conflict clusters". Furthermore, finding a minimum vertex set $V_{conf}$ whose removal renders a graph bipartite is an NP-Hard problem Yannakakis (1978). Our BFS-based approach serves as an efficient heuristic to approximate this set, which has proven extremely effective in practice. We therefore terminate the decomposition and assign all nodes in $V_{conf}$ a single, dedicated conflict color ($c = t$, where $t = 3$ in our framework). This avoids the computational overhead of a full recursive decomposition while still explicitly marking the most challenging regions for our downstream "Implicit Disambiguation" mechanism. The resulting $(t+1)$-value ground truth map, $Y_{Disco} \in \{0, ..., t\}^{H \times W}$, thus provides a supervisory signal that is both efficient and topologically rich. A deeper discussion on the theoretical and empirical bounds of CAG chromaticity is provided in Appendix A.4.

## 4.3 IMPLICIT DISAMBIGUATION: A DECOUPLED AND CONSTRAINED LOSS SYSTEM

While the "Explicit Marking" strategy provides a powerful supervisory signal, it has a theoretical limitation in its ambiguity within "secondary conflict" regions. To overcome this, we propose an "Implicit Disambiguation" mechanism, which resolves the inadequacies of discrete labels in the continuous feature space through a novel, end-to-end loss system. The total loss $\mathcal{L}_{total}$ is a weighted sum of five synergistic components:

$$\mathcal{L}_{total} = \mathcal{L}_{sem} + \mathcal{L}_{color} + \mathcal{L}_{cons} + \mathcal{L}_{conf} + \mathcal{L}_{adj} \tag{1}$$

The foundational semantic loss ($\mathcal{L}_{sem}$) and the weighted coloring loss ($\mathcal{L}_{color}$) supervise the primary semantic segmentation and instance classification tasks. A pair of complementary regularization terms—the consistency loss ($\mathcal{L}_{cons}$) and the conflict resolution loss ($\mathcal{L}_{conf}$)—act as a "push-pull" mechanism, respectively suppressing the misuse of the conflict color in simple bipartite regions and encouraging its precise prediction in topologically complex regions. Let $t$ be the class index for the conflict color, where $t = 3$ in our framework:

$$\mathcal{L}_{cons} = \mathbb{E}_{i \in M_{bip}}[(\sigma(P_{color}(i))_t)^2], \quad \mathcal{L}_{conf} = \mathbb{E}_{i \in M_{conf}}[(1 - \sigma(P_{color}(i))_t)^2] \tag{2}$$

Crucially, we designed the Adjacency Constraint Loss ($\mathcal{L}_{adj}$) as the key mechanism for resolving secondary conflicts. This loss operates on all adjacent edges in the graph, minimizing the cosine similarity between the mean probability vectors $\bar{P}(s)$ of neighboring instances, thus maximizing their orthogonality.

$$\mathcal{L}_{adj}(P_{color}, G) = \frac{1}{|E|} \sum_{(v_i, v_j) \in E} \frac{\bar{P}(s_i) \cdot \bar{P}(s_j)}{\|\bar{P}(s_i)\| \|\bar{P}(s_j)\|} \tag{3}$$

Conceptually, $\mathcal{L}_{adj}$ can be interpreted as a form of supervised contrastive loss. For each instance, other pixels within the same instance constitute the positive set, whose feature cohesion is encour-

aged by the classification loss $\mathcal{L}_{color}$. Conversely, all its adjacent instances in the graph are treated as explicit negative samples. By minimizing the cosine similarity to these negative samples, $\mathcal{L}_{adj}$ drives the learning of a feature manifold where the representations of different instances, especially adjacent ones, are maximally separated angularly. This achieves a powerful dual function: in bipartite regions, it drives the model to learn explicit, orthogonal 2-color encodings. In secondary conflict regions, it provides a counteracting gradient to the coloring loss, forcing the model to learn separable representations in the secondary feature dimensions. The effectiveness of this mechanism is significantly validated in our feature space visualization, as shown in Figure 6. A detailed formulation of each loss component is provided in Appendix A.5.

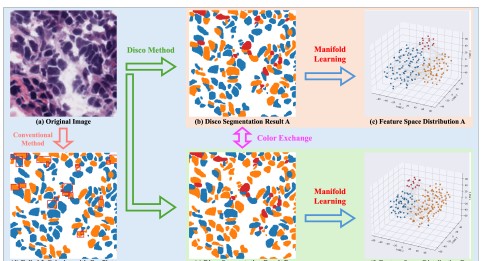

**Figure 6.** Visualization of implicit disambiguation and robustness to coloring ambiguity. We project the final 4D probability vector of each cell instance into a 3D space via t-SNE, visualizing the core "Implicit Disambiguation" mechanism of the Disco framework and its invariance to the non-uniqueness of coloring.

## 5 EXPERIMENTS

### 5.1 DATASETS

To comprehensively evaluate the performance and generalization capability of our Disco method, we conducted experiments on four publicly available and in-house datasets with significant heterogeneity. These datasets encompass diverse imaging modalities, cell types, and a wide spectrum of topological complexities. Prior to being processed by our pipeline, all datasets were preprocessed into non-overlapping $256 \times 256$ pixel patches. Specifically, the datasets used are PanNuke Gamper et al. (2020), DSB2018 Caicedo et al. (2019), CryoNuSeg Mahbod et al. (2021), and GBC-FS 2025.

**PanNuke** consists of 7,901 $256 \times 256$ images derived from H&E-stained, formalin-fixed paraffin-embedded (FFPE) tissue slides from 19 different human organs. It covers 5 distinct cell types and contains a total of 189,744 annotated instances. We adhere to its official data partitioning strategy, utilizing Fold 1 and Fold 2 for training and validation, and Fold 3 for final testing.

**DSB2018** comprises 670 fluorescence microscopy images of varying sizes (from $256 \times 256$ to $520 \times 696$), stained with DAPI and Hoechst. It includes a total of 29,443 annotated instances. We partitioned the dataset into 536 images for training, 67 for validation, and 67 for testing.

**CryoNuSeg** contains 30 H&E-stained frozen section images of size $512 \times 512$ from 10 human organs, with a total of 8,178 annotated instances. We cropped these images into $256 \times 256$ patches and subsequently partitioned them into 96 patches for training, 12 for validation, and 12 for testing.

**GBC-FS 2025**, first introduced in this study, is designed as a high-density case study to stress-test algorithmic robustness. Originating from a single, deeply annotated WSI of a gallbladder cancer frozen section, it provides an unprecedented challenge for handling complex topologies. It contains 2,839 $256 \times 256$ H&E-stained images with 864,204 annotated sub-cellular nuclei instances. The dataset was partitioned into 2,271 patches for training, 284 for validation, and 284 for testing. Further details are available in Appendix A.6.

### 5.2 IMPLEMENTATION DETAILS AND EVALUATION METRICS

All our experiments are conducted based on the PyTorch framework and run on a server equipped with eight NVIDIA RTX 4090 GPUs. We employ the Adam optimizer, with an initial learning rate set to $1 \times 10^{-4}$ and a weight decay of $5 \times 10^{-4}$. A step-wise learning rate decay strategy is adopted, which reduces the learning rate by a factor of 10 at the 70th epoch, supplemented by a linear warmup phase over the first 100 iterations to ensure initial training stability. All models are trained for a total of 200 epochs. Segmentation performance is evaluated using the Dice Coefficient (Dice) Lei et al. (2023), Aggregated Jaccard Index (AJI) Kumar et al. (2017), Detection Quality (DQ) Kirillov et al. (2019), Segmentation Quality (SQ), and Panoptic Quality (PQ) metrics. In all

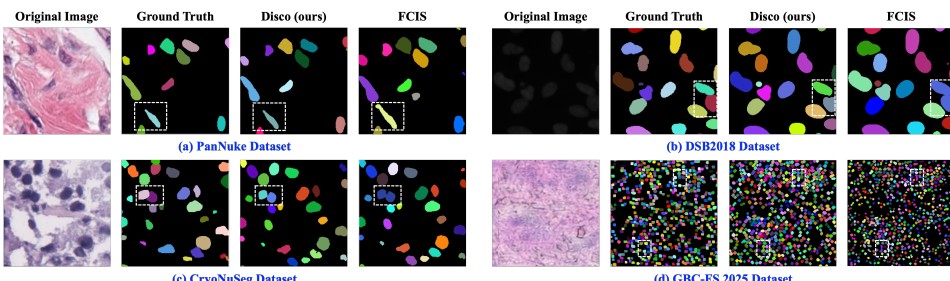

**Figure 7.** Qualitative comparison of segmentation results on the four datasets. Each panel displays (from left to right): the input image, the ground truth, the prediction from our Disco method, and the prediction from FCIS.

tables presented throughout this paper, the highest performance scores are highlighted in red bold, and the second-best scores are marked in blue bold.

## 5.3 EVALUATION AND RESULTS

Table 2: The comparison performances on Pan-Nuke dataset.

| Methods | Source | Metrics | | | | |
|---|---|---|---|---|---|---|
| | | DICE (↑) | AJI (↑) | DQ (↑) | SQ (↑) | PQ (↑) |
| DCAN | CVPR 2016 | 0.7785 | 0.5871 | 0.6591 | 0.7216 | 0.5065 |
| HoverNet | MIA 2019 | 0.7983 | 0.6463 | 0.7182 | 0.7824 | 0.5953 |
| NucleiSegNet | CBM 2021 | 0.7524 | 0.5446 | 0.6185 | 0.6892 | 0.4577 |
| DoNet | CVPR 2023 | 0.7812 | 0.6127 | 0.6849 | 0.7506 | 0.5445 |
| CPP-Net | IEEE TIP 2023 | 0.8145 | 0.6383 | 0.7115 | 0.7761 | 0.5831 |
| Un-SAM | MIA 2025 | 0.8017 | 0.6295 | 0.7043 | 0.7675 | 0.5702 |
| CellPose | NAT METHODS 2025 | 0.7871 | 0.6262 | 0.7035 | 0.7643 | 0.5918 |
| FCIS | ICML 2025 | 0.8185 | 0.6394 | 0.7252 | 0.8033 | 0.6109 |
| Disco | Ours | 0.8297 | 0.6566 | 0.7572 | 0.7943 | 0.6271 |

Table 3: The comparison performances on DSB2018 dataset.

| Methods | Source | Metrics | | | | |
|---|---|---|---|---|---|---|
| | | DICE (↑) | AJI (↑) | DQ (↑) | SQ (↑) | PQ (↑) |
| DCAN | CVPR 2016 | 0.7954 | 0.6764 | 0.7436 | 0.7807 | 0.6266 |
| HoverNet | MIA 2019 | 0.8983 | 0.7623 | 0.8637 | 0.8775 | 0.7625 |
| NucleiSegNet | CBM 2021 | 0.9045 | 0.6718 | 0.7845 | 0.8432 | 0.6821 |
| DoNet | CVPR 2023 | 0.8239 | 0.7162 | 0.7872 | 0.8294 | 0.6733 |
| CPP-Net | IEEE TIP 2023 | 0.9147 | 0.8131 | 0.8664 | 0.8793 | 0.7582 |
| Un-SAM | MIA 2025 | 0.9021 | 0.7863 | 0.8261 | 0.8347 | 0.7475 |
| CellPose | NAT METHODS 2025 | 0.9232 | 0.8247 | 0.8625 | 0.8715 | 0.7647 |
| FCIS | ICML 2025 | 0.9395 | 0.8287 | 0.8806 | 0.8789 | 0.7739 |
| Disco | Ours | 0.9454 | 0.8426 | 0.8753 | 0.8862 | 0.7781 |

**Quantitative and Qualitative Evaluation.** To validate the effectiveness and generalization capability of our Disco method, we conducted a comprehensive quantitative and qualitative comparison against a diverse set of mainstream instance segmentation methods across four heterogeneous datasets with varying topological complexities. The compared methods include the detection-based DoNet Jiang et al. (2023); contour prediction-based DCAN Chen et al. (2016) and NucleiSegNet Lal et al. (2021); distance mapping-based HoverNet Graham et al. (2019), CPP-Net Chen et al. (2023), and CellPose Stringer & Pachitariu (2025); the SAM-based foundation model Un-SAM Chen et al. (2025); and the graph-theory-based FCIS Zhang et al. (2025c). As demonstrated in Tables 2-5, our Disco method exhibits consistent and superior performance across all four datasets.

**Quantitative Comparison.** On the PanNuke dataset, which features the simplest topology with 100% bipartite cell adjacency graphs, our method achieves the best results on the key AJI and PQ metrics, with scores of 65.66% and 62.71%, respectively. This proves that even in the simplest cases, our dynamic coloring framework can automatically degenerate into a highly efficient 2-coloring model, outperforming the generic 4-color model FCIS which needs to learn a more complex representation. As the topological complexity increases, the advantages of Disco become even more pronounced. On the moderately complex DSB2018 and CryoNuSeg datasets, Disco also leads comprehensively in both AJI and PQ. It is worth noting that on DSB2018, although FCIS holds a slight edge in DQ, our method is significantly superior in SQ, indicating that our model generates more precise cell contours and ultimately prevails in the overall metrics.

This superiority culminates on the highly challenging GBC-FS 2025 dataset. This dataset, filled with dense "conflict clusters" and "secondary conflicts", poses a severe test for all methods. The results in Table 5 clearly show that Disco's performance on this dataset is overwhelming. Our method achieves an AJI of 52.09%, a remarkable 6.91% absolute improvement and a 15.3% relative improvement over the second-best method, FCIS. Concurrently, Disco achieves a comprehensive lead in both DQ and SQ dimensions. This irrefutably demonstrates that our "Explicit Marking + Implicit Disambiguation" strategy possesses unparalleled robustness and effectiveness when handling the extreme topological complexities of real-world scenarios.

**Visual Comparison.** We provide an intuitive visual comparison of the segmentation results in Figure 7. These visualizations showcase Disco's capability to accurately segment cell across the

Table 4: The comparison performances on Cry-oNuSeg dataset.

| Methods | Source | Metrics | | | | |
|---|---|---|---|---|---|---|
| | | DICE (↑) | AJI (↑) | DQ (↑) | SQ (↑) | PQ (↑) |
| DCAN | CVPR 2016 | 0.8649 | 0.5273 | 0.6184 | 0.7264 | 0.4682 |
| HoverNet | MIA 2019 | 0.8432 | 0.5585 | 0.6449 | 0.7543 | 0.4965 |
| NucleiSegNet | CBM 2021 | 0.8645 | 0.5764 | 0.6673 | 0.7841 | 0.5233 |
| DoNet | CVPR 2023 | 0.8594 | 0.5496 | 0.6584 | 0.7815 | 0.5147 |
| CPP-Net | IEEE TIP 2023 | 0.8784 | 0.5967 | 0.6742 | 0.7968 | 0.5379 |
| Un-SAM | MIA 2025 | 0.8627 | 0.5671 | 0.6594 | 0.7815 | 0.5567 |
| CellPose | NAT METHODS 2025 | 0.8869 | 0.5876 | 0.6861 | 0.8094 | 0.5724 |
| FCIS | ICML 2025 | 0.8977 | 0.5944 | 0.6929 | 0.8173 | 0.5793 |
| Disco | Ours | 0.9152 | 0.6134 | 0.7153 | 0.8329 | 0.5970 |

Table 5: The comparison performances on GBC-FS 2025 dataset.

| Methods | Source | Metrics | | | | |
|---|---|---|---|---|---|---|
| | | DICE (↑) | AJI (↑) | DQ (↑) | SQ (↑) | PQ (↑) |
| DCAN | CVPR 2016 | 0.7149 | 0.3983 | 0.5368 | 0.6572 | 0.3528 |
| HoverNet | MIA 2019 | 0.7394 | 0.4058 | 0.5468 | 0.6582 | 0.3699 |
| NucleiSegNet | CBM 2021 | 0.7419 | 0.4395 | 0.5618 | 0.6647 | 0.3835 |
| DoNet | CVPR 2023 | 0.7409 | 0.4198 | 0.5564 | 0.6705 | 0.3730 |
| CPP-Net | IEEE TIP 2023 | 0.7672 | 0.4311 | 0.5697 | 0.6765 | 0.3955 |
| Un-SAM | MIA 2025 | 0.7641 | 0.4406 | 0.5782 | 0.6907 | 0.4093 |
| CellPose | NAT METHODS 2025 | 0.7698 | 0.4376 | 0.5943 | 0.6749 | 0.4218 |
| FCIS | ICML 2025 | 0.7785 | 0.4518 | 0.5857 | 0.7068 | 0.4379 |
| Disco | Ours | 0.8137 | 0.5209 | 0.6795 | 0.7486 | 0.5087 |

four datasets, which vary significantly in staining, cell morphology, and density. In the challenging regions highlighted by white boxes, Disco demonstrates a clear advantage over the second-best method FCIS, successfully separating tightly clustered instances. More extensive visual results are presented in the Appendix A.7.

### 5.4 ABLATION STUDIES

To systematically validate the effectiveness of each core design within the Disco method, we conducted a series of exhaustive ablation studies on the most challenging GBC-FS 2025 dataset.

**Framework-level Comparison.** We first performed a framework-level comparison to establish the superiority of our dynamic coloring scheme, as shown in Table 6. The results clearly indicate that the pure 2-coloring baseline, which lacks a conflict resolution mechanism, performs the worst (37.85% AJI), further corroborating our earlier conclusion on the necessity of handling non-bipartite structures. Compared to the generic 4-coloring scheme of FCIS, our Disco framework achieves a significant 7.08% absolute improvement in the PQ metric. We infer that this advantage stems from our "divide and conquer" strategy, which provides the network with a clearer and more targeted learning objective, thereby avoiding the potential optimization difficulties and representation redundancy present in generic high-chromaticity models.

**Analysis of Loss Components.** We further dissected Disco's loss function system to verify the contribution of each component, as shown in Table 7. The experimental results demonstrate that every loss term we designed contributes positively to the final performance. Among them, the Adjacency Constraint Loss ($\mathcal{L}_{adj}$) shows a decisive influence; adding it alone to the basic Explicit Marking model yields a substantial 6% absolute improvement in PQ (from 42.57% to 48.26%). This result quantitatively and forcefully proves that our proposed $\mathcal{L}_{adj}$, aimed at resolving "secondary conflicts" through "Implicit Disambiguation", is the key driving force behind the success of the entire framework. When all loss components work in synergy, our final Disco model achieves the optimal performance of 50.87% in PQ, demonstrating that every component we designed is both necessary and effective.

Table 6: Ablation study on different coloring frameworks on GBC-FS 2025.

| Method | Coloring Scheme | Adjacency Constraint | Metrics | | |
|---|---|---|---|---|---|
| | | | Dice | AJI | PQ |
| Baseline | 2-Color | None | 0.7269 | 0.3785 | 0.3376 |
| FCIS | 4-Color | $L_{ort}$ on features | 0.7785 | 0.4518 | 0.4379 |
| Disco | Explicit Marking | $L_{adj}$ on probabilities | 0.8137 | 0.5209 | 0.5087 |

Table 7: Ablation study on the components of our Disco loss system on GBC-FS 2025.

| Method | $L_{cons}$ | $L_{conf}$ | $L_{adj}$ | Dice | AJI | PQ |
|---|---|---|---|---|---|---|
| Explicit Marking | × | × | × | 0.7543 | 0.4486 | 0.4257 |
| Explicit Marking | ✓ | ✓ | × | 0.7768 | 0.4711 | 0.4583 |
| Explicit Marking | × | × | ✓ | 0.7914 | 0.5057 | 0.4826 |
| Disco | ✓ | ✓ | ✓ | 0.8137 | 0.5209 | 0.5087 |

## 6 CONCLUSIONS

Through the first systematic analysis of the chromatic properties of cell adjacency graphs, we reveal the profoundly non-bipartite nature of real-world cell adjacency graphs, thereby elucidating the theoretical limitations of simple 2-coloring models in this domain. To address this challenge, we propose Disco, a novel framework operating on a "divide and conquer" principle. Disco employs an "Explicit Marking" strategy, which dynamically and intelligently decomposes the cell graph into a vast bipartite backbone and a sparse conflict set. More importantly, it achieves "Implicit Disambiguation" through a loss system featuring an end-to-end adjacency constraint. This mechanism resolves the discrete label ambiguities caused by "secondary conflicts" by learning separable representations in the continuous feature space. Comprehensive experiments demonstrate that Disco achieves state-of-the-art performance and excellent generalization across multiple heterogeneous datasets. Furthermore, we pioneer the use of the predicted "Conflict Map" as a novel interpretability tool, offering a new avenue for quantifying topological complexity. In summary, Disco provides a theoretically sound, efficient, and interpretable new paradigm for solving complex cell instance segmentation.

## 7 ACKNOWLEDGMENTS

This work was supported in part by the Shanghai Municipal Science and Technology Major Project (2023SHZDZX02 and 2017SHZDZX01 to L.J.), and in part by the Lingang Laboratory (Grant No. LGL-5555, LGL-6672 and LGL-8888). The computations in this research were performed using the CFFF platform of Fudan University.

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

## A APPENDIX

### A.1 LLM USAGE STATEMENT

In the preparation of this manuscript, we have strictly utilized the ChatGPT large language model (LLM) solely as a general auxiliary tool to support and enhance the writing process. All core scientific contributions, including the research concept, the design of all algorithms and experiments, and the final analysis of results, were exclusively developed by the human authors. LLMs were primarily employed to refine sentence structures for improved clarity and fluency, and to ensure terminology consistency. Additionally, an LLM-based image generation tool was used to create the logo graphic present in the paper's title. They made no contribution to any of the core research ideas or scientific conclusions presented in this paper.

### A.2 RELATED WORK

#### A.2.1 DETECTION-BASED INSTANCE SEGMENTATION

The "detect-then-segment" paradigm, introduced by pioneering works like Faster R-CNN Ren et al. (2016), was extended to instance segmentation by Mask R-CNN He et al. (2017). This two-stage method first generates bounding box Yi et al. (2019) proposals for individual instances and then performs segmentation within these localized regions. Its inherent capability for instance separation has made it a widely adopted framework, especially in semi-supervised cell segmentation tasks Zhou et al. (2020). However, the performance of this paradigm is fundamentally constrained by its reliance on axis-aligned bounding boxes, which provide a coarse representation for cells with complex or irregular morphologies. Furthermore, the final instance set is often dictated by the heuristic nature of non-maxima suppression (NMS), which frequently leads to the erroneous suppression of valid detections in dense, overlapping cell clusters, a critical failure mode we visualize in Figure 1(a).

#### A.2.2 CONTOUR AND DISTANCE-BASED SEGMENTATION

To circumvent the limitations of bounding boxes, a second major class of methods focuses on learning dense, pixel-wise representations that encode instance identity.

Contour-based methods aim to explicitly predict the boundaries between cells. The seminal U-Net Ronneberger et al. (2015) facilitated boundary learning by weighting cell edges in the loss function, with instances subsequently delineated via post-processing techniques like watershed. This architecture has profoundly influenced the field. Subsequent advancements focused on enhancing boundary prediction, for example, DCAN Chen et al. (2016) introduced dedicated boundary-class channels. Architectural optimizations such as nested skip connections in UNet++ Zhou et al. (2018) and multi-scale context aggregation in FullNet Qu et al. (2019) and CIA-Net Zhou et al. (2019) further improved boundary delineation. Despite these advances, contour-based methods remain highly sensitive to the binarization threshold of the predicted boundary map, leading to a trade-off between instance merging and fragmentation, as shown in Figure 1(b).

Distance-based methods achieve more robust separation by learning richer spatial relationships. StarDist Schmidt et al. (2018) and CellPose Stringer & Pachitariu (2025) learn to predict vectors from pixels to the cell's boundary or center, respectively. Hover-Net Graham et al. (2019) extended this concept by simultaneously predicting horizontal and vertical distance maps. Although these methods demonstrate SOTA performance, they typically rely on complex and error-prone post-processing Löffler et al. (2021) to reconstruct instances. This makes them susceptible to error propagation Chaudhary et al. (2021), where minute inaccuracies in the predicted vector fields can be

amplified into significant instance splitting errors, as illustrated in Figure 1(c). While recent models, from CPP-Net Chen et al. (2023) to Vision Transformer-based architectures like CellViT Hörst et al. (2024), continue to advance this paradigm, they still share a fundamental dependence on decoding local geometric cues.

### A.2.3 GRAPH-THEORETIC APPROACHES IN SEGMENTATION

A common theoretical bottleneck across all aforementioned paradigms is their lack of an intrinsic mechanism to model the global topological constraints among cells. To address this, a new perspective that abstracts the problem into a graph coloring task has emerged.

Graph coloring is a fundamental problem in graph theory with deep connections to scheduling and resource allocation. A $k$-coloring of a graph $G = (V, E)$ is a function $C : V \to \{1, ..., k\}$ such that for any edge $(u, v) \in E$, $C(u) \neq C(v)$. The minimum $k$ for which such a coloring exists is the graph's chromatic number, $\chi(G)$.

Early works in computer vision used graph coloring Nath et al. (2006) as a post-processing step for region merging. However, its integration into end-to-end learning is a recent innovation. Inspired by the Four-Color Theorem Gonthier et al. (2008), which states that any planar graph satisfies $\chi(G) \leq 4$, FCIS Zhang et al. (2025c) was the first to propose training a network to directly predict a 4-coloring of the cell graph. This pioneering work demonstrated the immense potential of reformulating instance segmentation as a multi-class classification problem constrained by topological rules.

A more fundamental concept than the Four-Color Theorem is 2-coloring, which, according to Kőnig's Theorem, is sufficient for any bipartite graph. This introduces a critical "Goldilocks" problem: what is the minimal, yet sufficient, number of colors required Kulikov et al. (2018) for real-world cellular topologies? Is the elegant simplicity of a 2-coloring model sufficient, or is the universal robustness of a 4-coloring model necessary?

Our work is dedicated to answering this core question and, based on the answer, proposing an optimal solution. We do not treat the cell graph as a passive object to which existing theories are applied; rather, we view it as a unique, unexplored structure. First, through systematic topological analysis, we are the first to provide an empirical answer to this question, revealing the complex reality of "near-bipartite but with dense conflict clusters." Second, based on this finding, we design a novel, non-iterative mechanism that injects strong topological priors into a deep network via a graph-aware loss system. Therefore, our work not only resolves the "Goldilocks" problem but also offers a new perspective on how to integrate profound graph-theoretic principles with deep representation learning to solve structured prediction tasks.

### A.3 SUPPLEMENTARY ANALYSIS AND VISUALIZATIONS

To provide a more holistic and intuitive understanding of our core motivations and the efficacy of our proposed coloring strategy, we present in Figure A.1 a comprehensive, side-by-side visual comparison across the full spectrum of dataset complexities. This figure visually decomposes our entire analysis pipeline, from the raw input to the final coloring outcomes, providing direct evidence for the key arguments made throughout this paper.

Column (a) is the input image, and Column (b) is the instance ground truth. These two columns establish the problem context, showcasing the significant heterogeneity in imaging modality (H&E, fluorescence), cell density, and morphology across the different benchmarks. The randomly colored instance map in (b) represents the ideal segmentation target.

Column (c) is the Cell Adjacency Graph (CAG) with odd-cycle detection. This column visualizes the core of our topological analysis. The instance map is abstracted into its underlying graph structure, where nodes represent cell centroids and edges represent adjacency. A critical diagnostic is performed on each graph: edges belonging to non-bipartite components are highlighted in red, while those of simple bipartite components remain gray. This visualization starkly reveals the "topological complexity spectrum": the graph for the PanNuke dataset is entirely gray, confirming its perfect bipartite nature. In DSB2018 and CryoNuSeg, red edges begin to appear, indicating the presence of isolated or small clusters of odd cycles. The graph for the GBC-FS 2025 dataset, in

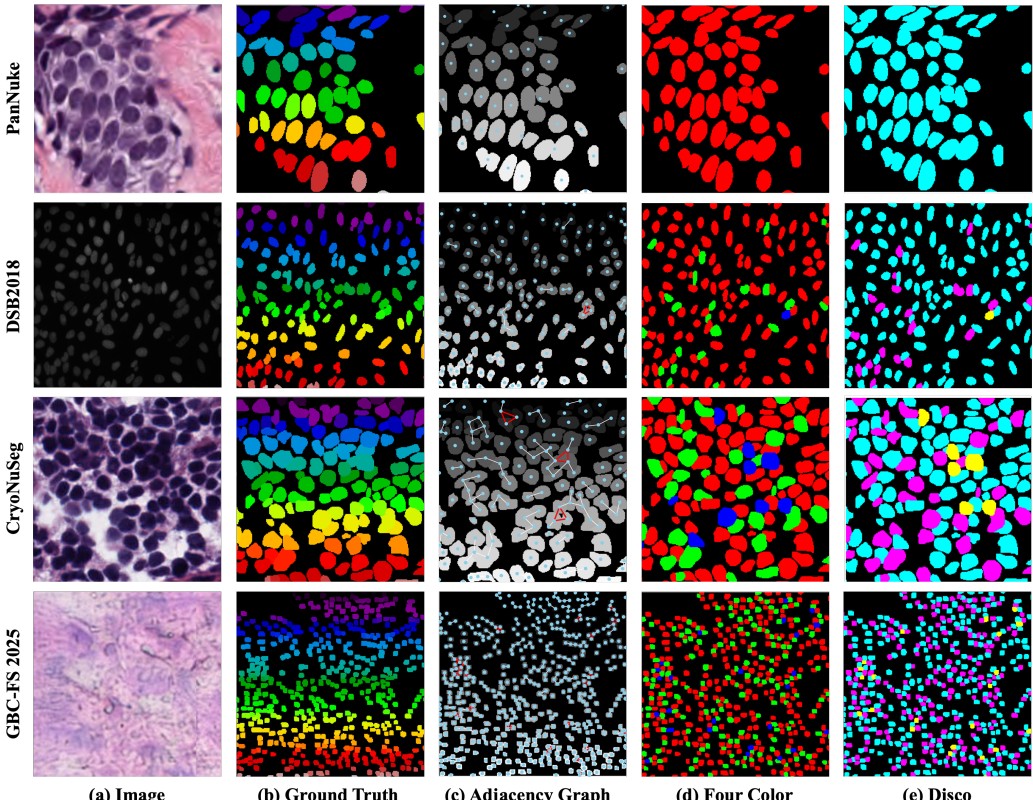

|  | (a) Image | (b) Ground Truth | (c) Adjacency Graph | (d) Four Color | (e) Disco |

**Figure A.1.** A comprehensive, cross-dataset visualization of cell graph topologies and coloring strategies. Each row corresponds to a specific dataset, with topological complexity increasing from top to bottom.

contrast, exhibits a massive, interconnected network of red conflict edges, visually demonstrating the prevalence of dense "conflict clusters" and "secondary conflicts".

Column (d) is the generic greedy 4-coloring. This column displays the results of applying a generic greedy coloring algorithm. A crucial empirical finding is revealed here: across all four datasets, including the highly complex GBC-FS 2025, a palette of three colors is empirically sufficient to resolve all adjacency constraints. This evidence strongly suggests that the true chromatic number of real-world cell graphs is typically $\chi(G) = 3$. Consequently, any framework based on the Four-Color Theorem, which prepares for a worst-case chromaticity of four, introduces unnecessary representational redundancy and potential optimization challenges.

Column (e) is the Disco coloring scheme. This final column showcases the adaptability and targeted efficiency of our proposed Disco framework. On the perfectly bipartite PanNuke dataset, our algorithm automatically degenerates into an optimal 2-coloring scheme, avoiding color waste. On the moderately complex DSB2018 and CryoNuSeg datasets, Disco effectively utilizes its conflict-aware 2-coloring strategy. It efficiently colors the vast bipartite backbones with two primary colors (blue and purple) and uses a dedicated conflict color (yellow) to precisely and sparsely mark the few topological conflicts identified in (c). On the extremely complex GBC-FS 2025, Disco robustly applies the same structured, conflict-aware coloring scheme. This provides a clear, topologically-informed supervisory signal that explicitly pinpoints the numerous conflict hotspots, laying the foundation for our "Implicit Disambiguation" mechanism to resolve the remaining ambiguities in the feature space.

In summary, this comprehensive visualization provides a powerful, side-by-side validation of our core thesis: Disco is not merely a coloring method, but a principled, data-driven framework that is tailor-made for the observed topological properties of real-world cell images. It avoids both the theoretical insufficiency of pure 2-coloring and the practical redundancy of 4-coloring, offering a solution that strikes an optimal balance between efficiency, robustness, and interpretability.

## A.4 Theoretical and Empirical Bounds on the Chromaticity of Cell Graphs

In Section 4.2, we justified our pragmatic choice to terminate the graph decomposition. This section provides a deeper theoretical and empirical discussion of the chromatic number ($\chi(G)$) of Cell Adjacency Graphs (CAGs), which further substantiates our design philosophy.

### A.4.1 Theoretical Remark on the Upper Bound of Chromaticity

A well-established result in graph theory provides an upper bound on the chromatic number based on the graph's maximum degree $\Delta_G$.

**Proposition 1**: For any graph $G$ with maximum degree $\Delta_G$, its chromatic number is bounded by $\chi(G) \leq \Delta_G + 1$.

**Proof**. This proposition can be proven constructively using a simple greedy coloring algorithm. The algorithm proceeds as follows:

1. Order the vertices of the graph $G$ arbitrarily as $v_1, v_2, ..., v_n$.

2. Provide a palette of $\Delta_G + 1$ available colors.

3. Iterate through the vertices from $v_1$ to $v_n$. For each vertex $v_i$, assign it the smallest available color that has not been used by any of its already-colored neighbors in $\{v_1, ..., v_{i-1}\}$.

This algorithm is guaranteed to succeed. When coloring any vertex $v_i$, it has at most $\Delta_G$ neighbors. Therefore, at most $\Delta_G$ colors could have been used by its neighbors. Since our palette contains $\Delta_G + 1$ distinct colors, there will always be at least one color available for $v_i$. As this holds for all vertices, the entire graph can be colored with at most $\Delta_G + 1$ colors, thus establishing the bound $\chi(G) \leq \Delta_G + 1$. This bound is related to the more powerful Brooks' Theorem Kim (1995), which provides a tighter bound of $\chi(G) \leq \Delta_G$ for most graphs.

Our data analysis (Table 1) shows that for our datasets, $\Delta_G \leq 8$, implying a worst-case theoretical bound of 9 colors based on this proposition. This loose upper bound highlights a significant gap between worst-case graph theory and the specific reality of our problem domain.

### A.4.2 Empirical Findings: The "3-Color Phenomenon"

Our most striking empirical finding, as illustrated in the qualitative results (e.g., Figure A.1 in Appendix A.2), is that a palette of three colors is consistently sufficient to resolve all adjacency constraints across all four diverse datasets, including the topologically complex GBC-FS 2025. This suggests that the true chromatic number of real-world cell graphs is almost always $\chi(G) = 3$ if they are non-bipartite, and $\chi(G) = 2$ otherwise. This "3-Color Phenomenon" stands in stark contrast to the theoretical upper bound of 9.

### A.4.3 A Hypothesis on the Structural Origins of Low Chromaticity in Cell Graphs

The consistent observation of low chromaticity motivates a deeper question: why are cell graphs so structurally constrained? We hypothesize that this is not a coincidence, but a consequence of fundamental physical and biological principles governing tissue organization.

Graph theory tells us that a high chromatic number ($\chi(G) \geq 4$) requires the graph to contain a $K_4$ minor—that is, a subgraph that can be contracted to form a complete graph of 4 vertices ($K_4$). A $K_4$ subgraph represents a scenario where four distinct cell instances are all mutually adjacent to each other, as shown in Figure A.2.

We argue that such a configuration is biophysically improbable in a quasi-2D tissue slice. Due to constraints of cell volume, membrane tension, and optimal packing principles (analogous to circle packing problems), it is extremely difficult for four distinct, roughly convex objects to all achieve simultaneous, pairwise physical contact. While three-way contact (forming a 3-cycle or triangle) is common at intercellular junctions, a four-way mutual contact is a topologically unstable and energetically unfavorable state.

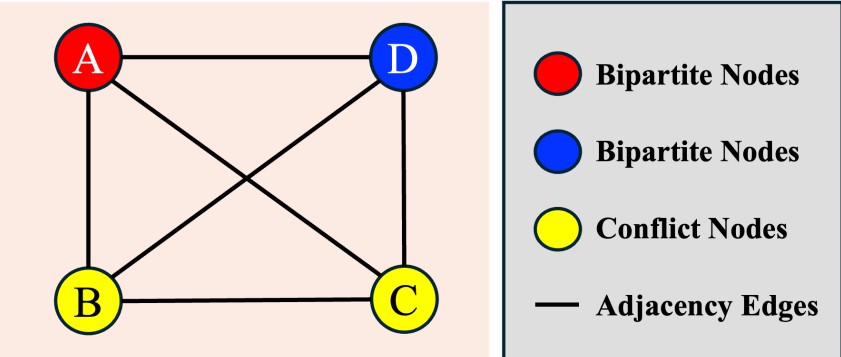

**Figure A.2.** A schematic of a complete graph on four vertices ($K_4$) and its biophysical interpretation in cell segmentation. A $K_4$ subgraph represents a topological configuration where four distinct cell instances (A, B, C, D) are all mutually adjacent. We hypothesize that forming such a structure in a quasi-2D tissue slice is biophysically improbable due to the physical constraints of cell volume and membrane tension. The absence of such high-order conflict structures like $K_4$ minors in real-world cell graphs provides a theoretical underpinning for the empirically observed "3-Color Phenomenon".

Conclusion and Justification for Disco: This hypothesis provides a theoretical underpinning for our empirical "3-Color Phenomenon". It suggests that the underlying biophysics of cellular arrangements naturally precludes the formation of high-order conflict structures (like $K_4$ minors) that would necessitate 4 or more colors. This finding is the ultimate justification for our Disco framework. Instead of preparing for a high-chromaticity worst case that is biophysically unrealistic (like a 4-coloring scheme), our 2+1 approach is perfectly tailored to the observed reality: it efficiently handles the dominant bipartite structures with two colors and reserves a single, sufficient conflict color to resolve the common, but low-order (predominantly 3-cycle), topological conflicts.

## A.5 Detailed Formulation of the Loss System

The end-to-end optimization of our Disco framework is driven by a decoupled and constrained loss system. The total loss, as introduced in Equation (1), is a weighted sum of five components. Here, we provide a detailed formulation and rationale for each component.

**Notation**: Let $P_{sem} \in \mathbb{R}^{B \times 2 \times H \times W}$ be the semantic logits and $P_{color} \in \mathbb{R}^{B \times (t+1) \times H \times W}$ be the coloring logits produced by the network for a batch of $B$ images. Let $Y_{sem} \in \{0, 1\}^{B \times H \times W}$ be the binary semantic ground truth, and $Y_{Disco} \in \{0, ..., t\}^{B \times H \times W}$ be our Disco ground truth map, with $t = 3$ in our case. Let $\sigma(\cdot)$ be the channel-wise Softmax function. Let $\Omega$ denote the set of all pixel coordinates in an image.

**1. Foundational Losses $\mathcal{L}_{sem}$ and $\mathcal{L}_{color}$**

These two losses provide the primary supervisory signal for the network's dual heads.

The Semantic Loss $\mathcal{L}_{sem}$ supervises the basic foreground/background segmentation. It is a linear combination of a pixel-wise Cross-Entropy loss $\mathcal{L}_{CE}$ and a region-based Dice loss $\mathcal{L}_{Dice}$:

$$\mathcal{L}_{sem}(P_{sem}, Y_{sem}) = \mathcal{L}_{CE}(P_{sem}, Y_{sem}) + \mathcal{L}_{Dice}(P_{sem}, Y_{sem}) \tag{4}$$

This ensures the network learns to accurately identify the spatial support of all cellular regions.

The Coloring Loss $\mathcal{L}_{color}$ supervises the multi-class Disco labeling task. It is composed of a Weighted Cross-Entropy $\mathcal{L}_{WCE}$ and a Dice loss:

$$\mathcal{L}_{color}(P_{color}, Y_{Disco}) = \mathcal{L}_{WCE}(P_{color}, Y_{Disco}) + \mathcal{L}_{Dice}(P_{color}, Y_{Disco}) \tag{5}$$

The $\mathcal{L}_{WCE}$ is defined for a pixel $i$ as:

$$\mathcal{L}_{WCE}(P_{color}(i), Y_{Disco}(i)) = -w_{Y_{Disco}(i)} \log(\sigma(P_{color}(i))_{Y_{Disco}(i)}) \tag{6}$$

where $w_c$ is the weight for class $c$. We set the weight for the conflict class $w_t$ significantly higher than for the bipartite colors ($w_t \gg w_{c \in \{1,...,t-1\}}$) to compel the model to prioritize learning these critical topological features.

**2. Decoupled Regularization Losses $\mathcal{L}_{cons}$ and $\mathcal{L}_{conf}$**

To enforce the specific logic of our "divide and conquer" scheme, we introduce a pair of complementary regularization losses that act as a "push-pull" mechanism.

The Bipartite Consistency Loss $\mathcal{L}_{cons}$ acts as the "pull" force. It penalizes the erroneous prediction of the conflict color within the topologically simple bipartite regions, $M_{bip} = \{i \in \Omega \mid Y_{Disco}(i) \in \{1,...,t-1\}\}$. Its objective is to ensure the sparsity and targeted use of the conflict class.

$$\mathcal{L}_{cons}(P_{color}, M_{bip}) = \mathbb{E}_{i \in M_{bip}}[(\sigma(P_{color}(i))_t)^2] \tag{7}$$

Conversely, the Conflict Resolution Loss $\mathcal{L}_{conf}$ acts as the "push" force. It rewards high-confidence predictions for the conflict class within the regions of topological conflict, $M_{conf} = \{i \in \Omega \mid Y_{Disco}(i) = t\}$, ensuring these crucial signals are not suppressed.

$$\mathcal{L}_{conf}(P_{color}, M_{conf}) = \mathbb{E}_{i \in M_{conf}}[(1 - \sigma(P_{color}(i))_t)^2] \tag{8}$$

**3. Adjacency Constraint Loss $\mathcal{L}_{adj}$ for Implicit Disambiguation**

This is the cornerstone of our "Implicit Disambiguation" mechanism, designed to resolve the theoretical limitations of discrete labels in secondary conflict regions. It operates on the continuous feature manifold of the network's output, transforming an ill-posed discrete problem into a well-posed feature separation problem.

Let $s_k$ denote the set of pixel coordinates for instance $k$. We first define the mean probability vector for this instance, $\bar{P}(s_k) \in \mathbb{R}^{t+1}$, as the average of the softmax probability vectors over all its pixels:

$$\bar{P}(s_k) = \mathbb{E}_{p \in s_k}[\sigma(P_{color}(p))] \tag{9}$$

This vector represents the instance's average feature embedding in the probability space.

The Adjacency Constraint Loss ($\mathcal{L}_{adj}$) is then defined as a global potential function over all edges $E$ in the Cell Adjacency Graph $G$. It aims to maximize the angular separation between the feature representations of adjacent instances by minimizing their cosine similarity.

$$\mathcal{L}_{adj}(P_{color}, G) = \frac{1}{|E|} \sum_{(v_i, v_j) \in E} \frac{\bar{P}(s_i) \cdot \bar{P}(s_j)}{\|\bar{P}(s_i)\|_2 \cdot \|\bar{P}(s_j)\|_2 + \epsilon} \tag{10}$$

where $\| \cdot \|_2$ denotes the L2 norm (Euclidean norm) of a vector, and $\epsilon$ is a small constant $10^{-8}$ for numerical stability.

This loss provides a powerful counteracting gradient to the primary classification objective. In secondary conflict regions, where $\mathcal{L}_{color}$ drives two adjacent instances towards the same discrete class $t$, $\mathcal{L}_{adj}$ forces their continuous representations apart by compelling the network to learn different activation patterns in the secondary channels ($c \in \{1,...,t-1\}$). Conceptually, this can be interpreted as a form of supervised contrastive loss, where the graph adjacency defines negative pairs that must be repelled in the feature space. This mechanism ensures that even instances with ambiguous discrete labels become separable on the learned feature manifold, which is critical for accurate instance reconstruction in topologically complex regions.

## A.6 THE GBC-FS 2025 HIGH-DENSITY CASE STUDY DATASET

This section provides a comprehensive description of the GBC-FS 2025 (GallBladder Cancer Frozen Section 2025) dataset, which was constructed and first introduced in this study. We detail the clinical motivation, data curation process, key characteristics, and its potential impact on the field.

### A.6.1 THE CRITICAL GAP IN HIGH-DENSITY PATHOLOGY SEGMENTATION

Gallbladder cancer (GBC) is the most common biliary tract malignancy, accounting for 80–95% of cases, and ranks among the major digestive system cancers. Due to its insidious onset and lack of specific early symptoms, most patients are diagnosed at an advanced stage, resulting in extremely poor prognosis: the 5-year survival rate for metastatic disease is below 5%, with a median survival often under 6 months. This reality highlights the urgent need for high-quality, cell- and subcellular-level quantitative analysis to enable breakthroughs in early pathological feature identification, immune infiltration analysis, and spatial heterogeneity characterization.

However, the histological features of GBC pose formidable challenges. Tumor cells often form compact nests with ambiguous boundaries, while immune responses induce dense lymphocyte clusters and tertiary lymphoid structures (TLS) at the tumor periphery, creating regions of extreme density and overlap. In these areas, conventional cell segmentation approaches frequently fail, leading to downstream biases such as distorted immune infiltration assessment and unreliable biomarker discovery.

Existing public datasets (e.g., MoNuSeg, PanNuke, Lizard/CoNIC), though valuable, are centered on more common cancers and lack sub-nuclear granularity. As a result, mainstream models generalize poorly to highly dense and complex tissues like GBC. The GBC-FS 2025 dataset was created to explicitly fill this gap, providing the first large-scale, sub-nuclear annotated benchmark on frozen sections of gallbladder cancer.

### A.6.2 DATASET CURATION AND SUB-CELLULAR ANNOTATION

**(1) Data Source**: The 2,839 $256 \times 256$ images in the GBC-FS 2025 dataset originate from a single whole-slide image (WSI) of a gallbladder cancer frozen section, obtained from a 71-year-old female with Stage IVB adenocarcinoma. Frozen sections inherently exhibit uneven staining, ice-crystal artifacts, tissue distortion, and blurred or incomplete nuclear membranes, making reliable cell-level nuclear annotation infeasible. Therefore, instead of annotating individual cells, which cannot be consistently identified under such conditions, we focus on annotating the chromatin-like dense sub-cellular aggregates that remain visually stable and reproducible. The complete WSI data of the remaining patients will also be released later.

**(2) Annotation Methodology**: All instances were manually delineated by four annotators with basic morphological training using QuPath (v0.5.1) software, and were reviewed by a Ph.D. with a pathology background. For regions with significant ambiguity, a pathologist was consulted for limited sample verification to confirm that the overall annotation direction was consistent with the basic morphological features of frozen sections. Additionally, our annotation target was "chromatin-like dense sub-cellular structures" as visual primitives, as these structures are more consistently identifiable in noisy frozen sections than complete nuclei. This was a deliberate choice aimed at creating a benchmark that reflects the true visual challenges of this data modality, rather than approximating cleaner, FFPE-style annotations.

**(3) Quality Control**: Given the inherent ambiguity of boundaries in frozen sections, we adopted a multi-round expert review process, which is more suitable for this scenario than traditional IoU/Dice metrics. Annotations were completed by junior annotators and then reviewed by a Ph.D. with a pathology background. Any uncertain regions were marked and then adjudicated by senior team members, with a few difficult cases submitted to a pathologist for sample inspection.

**(4) Tissue Composition**: A pathologist-guided regional assessment indicates that the dataset is composed of approximately 89.69% cancer cell regions, 6.11% fibroblast regions, and 4.20% immune cell regions.

The dataset carries practical significance on three fronts. Biologically, it facilitates exploration of dividing nuclei and nuclear fragmentation, enabling quantification of heterogeneity and multipolarity

Table 8: Overview of the datasets used in our experiments. FFPE denotes formalin-fixed paraffin-embedded, while FS denotes frozen section.

| Dataset | Imaging Modality | Tissue Processing | Total Patches | Total Annotated Instances | Avg. Cells per Patch |
|---|---|---|---|---|---|
| PanNuke | H&E, Bright-field | FFPE | 7901 | 189,744 | 20.92 |
| DSB2018 | Fluorescence | FFPE | 670 | 29,443 | 43.94 |
| CryoNuSeg | H&E, Bright-field | FS | 120 | 8,178 | 68.15 |
| GBC-FS 2025 | H&E, Bright-field | FS | 2839 | 864,204 | 304.44 |

that inform tumor aggressiveness and prognosis. Algorithmically, sub-nuclear annotations provide finer-grained supervisory signals than traditional cell-level labels, enhancing robustness in dense, overlapping, and morphologically complex regions, and serving as an ideal dataset for stress-testing segmentation models. Clinically, sub-nuclear delineations not only benchmark nucleus segmentation but also inform cell-segmentation preprocessing, as aggregating nuclear units can approximate whole-cell boundaries, thereby supporting immune infiltration analysis, tumor grading, and prognosis assessment.

### A.6.3 KEY CHARACTERISTICS AND COMPARISON WITH EXISTING DATASETS

GBC-FS 2025 is distinguished from existing datasets not only by its focus on a rare and challenging cancer type but, more importantly, by its unprecedented topological complexity and sub-nuclear annotation granularity. To contextualize its unique properties, we provide a direct comparison with the other datasets used in this study in Table 8.

As the table clearly illustrates, GBC-FS 2025 sits at the extreme end of the "topological complexity spectrum." It exhibits by far the highest average instance density, with 304.44 instances per patch—more than four times that of the next densest dataset, CryoNuSeg. This extreme density is the primary driver of its complex graph structures, which we analyzed in the main paper (Section 3). A staggering 30.49% of its cell nodes are classified as conflict nodes, and 24.64% are involved in secondary conflicts, figures that are an order of magnitude higher than those of other public benchmarks.

This unique combination of massive scale (over 860k instances) and extreme topological complexity makes GBC-FS 2025 an indispensable resource for the field. It serves as a crucial complement to existing datasets by providing a much-needed benchmark for the most challenging, high-density segmentation scenarios, enabling the development and rigorous stress-testing of robust, topology-aware algorithms like our proposed Disco.

### A.7 SUPPLEMENTARY QUALITATIVE RESULTS

To provide a more comprehensive visual testament to the robustness and generalization capability of our Disco framework, we present in Figure A.3 an extensive gallery of qualitative segmentation results across the four datasets. This figure showcases Disco's performance in a wide variety of challenging scenarios, highlighting its consistent accuracy across different imaging modalities, staining techniques, cell morphologies, and densities.

These visualizations demonstrate Disco's remarkable ability to adapt to vastly different visual domains, performing equally well on bright-field H&E-stained images and on fluorescence microscopy images. Furthermore, it successfully handles the entire spectrum of cell densities, from sparse arrangements to extremely dense and cluttered environments. In highly challenging regions, Disco consistently generates instance masks that are highly concordant with the ground truth, accurately delineating irregular shapes and separating tightly clustered cells.

A particularly noteworthy observation is showcased in cases with known incomplete annotations. Our Disco method, trained only on the available labels, exhibits a powerful generalization capability: it successfully identifies and segments numerous sub-cellular nuclei that were entirely absent from the ground truth annotation. This can be clearly observed by comparing the Disco (ours) row with the Ground Truth row in the corresponding panels, where our method correctly delineates many additional, valid instances. This "gap-filling" ability underscores the robustness of the learned topological priors and highlights Disco's future potential in semi-supervised or weakly-supervised

Table 9: Computational cost comparison. Params denotes model parameters. FLOPs are calculated for a $256 \times 256$ input. FPS denotes frames per second during inference.

| Method | Backbone | Source | Paras | FLOPs | FPS |
|--------|----------|--------|-------|-------|-----|
| HoverNet | ResNet-50 | MIA 2019 | 49.70M | 192.70G | 12.4 |
| FCIS | Swin-Unet | ICML 2025 | 48.23M | 66.37G | 28.5 |
| Disco | Swin-Unet | Ours | 46.84M | 65.21G | 29.2 |

learning scenarios, significantly enhancing its practical utility in real-world applications where exhaustive annotation is often infeasible.

## A.8   COMPARISON OF COMPUTATIONAL EFFICIENCY

Our analysis confirms that the graph-based components of our method (CAG construction, BFS labeling) are one-time, offline preprocessing steps and introduce no additional overhead during inference. The inference speed is therefore determined solely by the neural network's architecture and post-processing.

To ensure a fair comparison of inference efficiency, we benchmarked all models on a single NVIDIA RTX 4090 GPU with $256 \times 256$ inputs. The results are summarized in Table 9.

The results clearly demonstrate the high efficiency of our Disco framework. Disco achieves an inference speed of 29.2 FPS, which is not only highly competitive but also slightly faster than the strong graph-based baseline, FCIS (28.5 FPS). Furthermore, it is more than twice as fast as methods like HoverNet that rely on computationally intensive post-processing.

Notably, even with its sophisticated dual-branch head and decoupled loss system, our Disco model is more parameter-efficient than FCIS (46.84M vs. 48.23M) and has a lower computational load (65.21G vs. 66.37G FLOPs). These results irrefutably demonstrate that Disco achieves its state-of-the-art accuracy not by adding significant computational complexity, but through a more efficient and targeted learning strategy. Our method successfully improves performance without sacrificing, and in some cases even improving upon, practical computational efficiency.

## A.9   COMPARISON BETWEEN $\lambda_{adj}$ AND NONADJLOSS

While NonAdjLoss Ganaye et al. (2019) and our $\lambda_{adj}$ share a conceptual similarity—both leverage a predefined adjacency graph to penalize undesired relationships in the network's output—their objectives, mechanisms, and application domains are fundamentally different, which in turn highlights the uniqueness of our method.

**(1) Fundamentally Different Objectives.** The objective of NonAdjLoss is semantic-level correctness, aimed at enforcing anatomical consistency by preventing impossible adjacencies between different semantic classes. It is a loss that promotes regional purity and semantic correctness. In contrast, the objective of our $\lambda_{adj}$ is instance-level discriminability, aimed at enforcing feature dissimilarity between different instances within the same semantic class. The former promotes regional purity, while the latter promotes instance separability.

**(2) Different Operational Mechanisms.** NonAdjLoss operates at the pixel level, penalizing the simultaneous occurrence of forbidden class probabilities within a local neighborhood. Our $\lambda_{adj}$, however, is a form of supervised contrastive loss that operates at the instance level. It first abstracts each instance into a mean feature vector, then treats the edges in the adjacency graph as defining "negative pairs," and directly minimizes the feature similarity of these pairs. This mechanism of enforcing the separation of the feature manifold at the instance level, based on contrastive learning, is not present in NonAdjLoss.

**(3) Distinct Application Domains and Problems Solved.** NonAdjLoss addresses the problem of "topologically incorrect semantic layouts." Our $\lambda_{adj}$ addresses the problem of "ambiguous instance identity in dense clusters," particularly the "secondary conflict" challenge, which is a unique issue in graph-coloring-based instance segmentation.

In summary, although both methods use an adjacency graph, they are applied to solve problems of two entirely different dimensions and employ fundamentally distinct mechanisms. NonAdjLoss

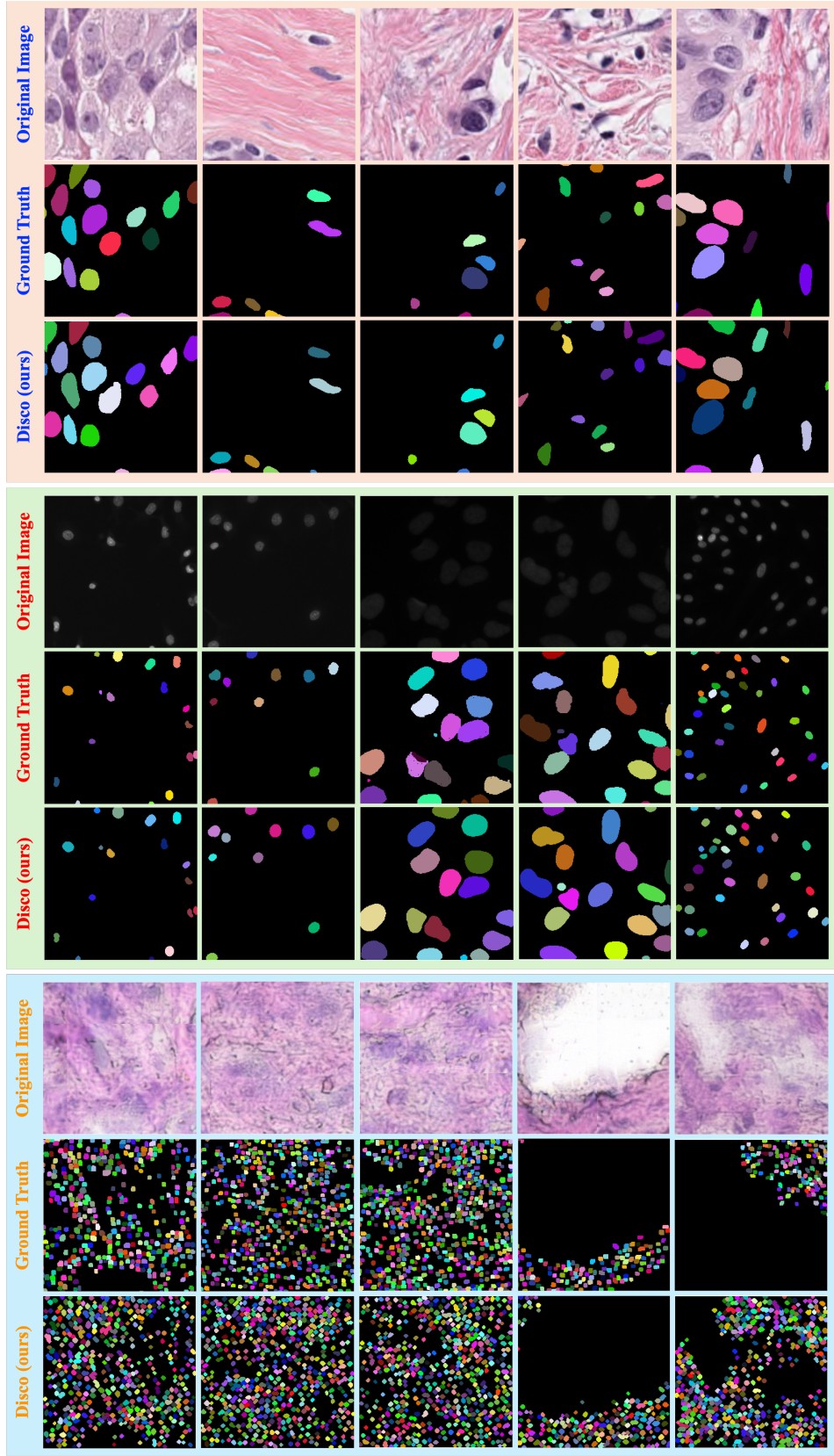

**Figure A.3.** A gallery of qualitative segmentation results from our Disco method on the four datasets. For each dataset block, the top row displays the original input images, the middle row shows the corresponding ground truth instance masks, and the bottom row presents our Disco's final segmentation predictions. For clarity, all instance masks are visualized with random colors.

Table 10: Fair comparison of different coloring strategies at GBC-FS 2025.

| Method | DICE | AJI | DQ | SQ | PQ |
|---|---|---|---|---|---|
| Greedy 3-Coloring | 0.7528 | 0.4594 | 0.5838 | 0.6971 | 0.4122 |
| FCIS | 0.7785 | 0.4518 | 0.5857 | 0.7068 | 0.4379 |
| **Disco** | **0.8137** | **0.5209** | **0.6795** | **0.7486** | **0.5087** |

ensures that different classes do not touch when they should not, whereas our $\lambda_{adj}$ ensures that different instances of the same class have separable features when they do touch.

### A.10 FAIR COMPARISON OF DIFFERENT COLORING STRATEGIES

Our (2+1) scheme, while seemingly creating an intermediate ambiguity, constitutes a superior, more efficient, and more "dynamically adaptive" learning paradigm compared to a static, greedy 3-coloring that always uses three colors. Our rationale is twofold.

(1) **It is a dynamically adaptive and more efficient labeling strategy.** A standard greedy 3-coloring algorithm will always attempt to use three colors, regardless of whether the input graph is simple or complex. In contrast, our "Explicit Marking" algorithm is dynamically adaptive. When faced with a simple, bipartite cell graph (such as in the PanNuke dataset), our algorithm automatically degenerates into an optimal 2-coloring scheme, using only two colors and completely avoiding the introduction of a third. It only "activates" and introduces the dedicated conflict color when it detects an unavoidable topological conflict in the graph. This "on-demand allocation" strategy is not only more elegant in theory but also provides the network with a more concise supervisory signal that adheres to the "Occam's razor" principle, avoiding representational redundancy in simple scenarios.

(2) **It drives more robust feature learning.** The "label ambiguity" is a deliberate design choice that, when coupled with our $\lambda_{adj}$, creates a constructive tension that drives the learning of a more robust feature manifold. A clean 3-coloring supervision would only require the network to learn to mimic discrete labels. Our (2+1) scheme, however, compels the model to resolve the conflict between the coloring loss and the adjacency loss by utilizing secondary feature dimensions, thereby learning a truly separable, topologically-aware feature representation.

To empirically validate this core argument, we conducted a relevant ablation study. We performed an experimental validation on the GBC-FS 2025 dataset, comparing our Disco method with a standard greedy 3-coloring method and the FCIS method. The results, as shown in Table 10, demonstrate that our Disco framework, with its "Explicit Marking + Implicit Disambiguation" strategy, significantly outperforms the models trained on clean, unambiguous 3-color and 4-color labels. This substantial performance gap provides strong empirical evidence that our seemingly "roundabout" design is, in fact, a more effective learning strategy that guides the model to a more robust and precise solution.

