# OpenReview forum: "Disco: Densely-overlapping Cell Instance Segmentation via Adjacency-aware Collaborative Coloring"
_ICLR.cc/2026/Conference — ICLR 2026 Poster_

### Official Review · Reviewer_SvPT · 2025-10-18

**Soundness:** 4
**Presentation:** 3
**Contribution:** 4
**Rating:** 8
**Confidence:** 5

**Summary:**

This paper presents DISCO, an effective framework for dense cell instance segmentation with detailed theoretically analysis. The authors identify the lack of global topological awareness in existing methods, and provide solution by graph coloring theory.
Through the introduction of a new GBC-FS 2025 dataset and a systematic cross-dataset topological analysis, the authors show that cell adjacency graphs are predominantly non-bipartite with a high density of odd-length cycles.  DISCO addresses the challenge by the proposed explicit marking and implicit disambiguation mechanism.
Comprehensive experiments across four datasets demonstrate consistent and significant gains, with qualitative visualizations and detailed ablations further verify Disco's effectiveness.

**Strengths:**

1.The authors construct a large-scale GBC-FS 2025 benchmark which contains highly complex and dense sub cellular structures. The also report the comparisons of recent models on the benchmark.

2.The paper conduct systematic, quantitative analysis of the complex topology of cell adjacency graphs and reveal their inherent non-bipartite nature. They also establish a clear conceptual shift from local geometric modeling to global topological reasoning for instance segmentation.

3.The “divide and conquer” strategy handles bipartite regions efficiently while explicitly modeling non-bipartite conflict clusters. The integration of “Explicit Marking” and “Implicit Disambiguation” aligns well align with the theoretical analysis.

**Weaknesses:**

1. The method introduces multiple loss components and graph-based computations. Although training settings are reported, there is limited discussion or empirical evidence regarding runtime and memory overhead compared to baselines such as FCIS or HoverNet. This omission is particularly notable since the paper highlights DISCO’s efficiency advantage over FCIS.

2. While the topological observations are empirically sound, the paper stops short of providing formal theoretical bounds or proofs. Incorporating theoretical justification such as convergence analysis or expected bounds would enhance the rigor and clarity of the claims.

3. It would be beneficial to discuss or empirically compare against recent GNN-based segmentation or relational reasoning approaches. This would help contextualize the novelty and positioning of the proposed framework within the broader landscape of graph-based learning methods.

4. Although the paper provides an anonymous repository, the released materials currently lack full access to the proposed GBC-FS dataset and the complete implementation of DISCO. This incomplete release raises concerns about reproducibility. It would be helpful to clarify whether the dataset and code will be fully released upon publication, as stated in the reproducibility section.

**Questions:**

1. Could the authors provide quantitative evidence (e.g., runtime, GPU memory usage) to support the claim that DISCO is more efficient than FCIS, particularly given its additional loss terms and graph computations?

2. Do the authors plan to fully open-source the GBC-FS dataset and the complete DISCO implementation after the review phase?

3. Can the authors offer theoretical insights or proofs regarding the efficiency or convergence of DISCO?

4. Have the authors considered comparing DISCO to GNN-based instance segmentation or relational reasoning methods?

**Details Of Ethics Concerns:**

N.A.

---

> ### Author Response · Authors · 2025-11-21
> **We thank the reviewer for their comments and suggestions. We will address the questions raised one by one and make corresponding modifications or additions in the revised version of our paper.**
>
> # 1. Regarding the comparison of computational efficiency
>
> We thank the reviewer for this important practical consideration. **We have conducted a comprehensive analysis of the computational cost of Disco and key baselines, and will add a detailed “Computational Cost” comparison table to the Appendix.**
>
> Our analysis confirms that the graph-based components of our method (CAG construction, BFS labeling) are one-time, offline preprocessing steps and introduce no additional overhead during inference. The inference speed is therefore determined solely by the neural network's architecture and post-processing.
>
> To ensure a fair comparison of inference efficiency, we benchmarked all models on a single NVIDIA RTX 4090 GPU with $256\times256$ inputs. The results are summarized in Table R12.
>
> **Table R12**: Computational cost comparison. Params denotes model parameters. FLOPs are calculated for a $256\times256$ input. FPS denotes frames per second during inference.
>
> | Method   | Backbone  | Source    | Paras  | FLOPs   | FPS  |
> | -------- | --------- | --------- | ------ | ------- | ---- |
> | HoverNet | ResNet-50 | MIA 2019  | 49.70M | 192.70G | 12.4 |
> | FCIS     | Swin-Unet | ICML 2025 | 48.23M | 66.37G  | 28.5 |
> | Disco    | Swin-Unet | Ours      | 46.84M | 65.21G  | 29.2 |
>
> **The results clearly demonstrate the high efficiency of our Disco framework.** Disco achieves an inference speed of 29.2 FPS, which is not only highly competitive but also slightly faster than the strong graph-based baseline, FCIS (28.5 FPS). Furthermore, it is more than twice as fast as methods like HoverNet that rely on computationally intensive post-processing.
>
> Notably, even with its sophisticated dual-branch head and decoupled loss system, our Disco model is more parameter-efficient than FCIS (46.84M vs. 48.23M) and has a lower computational load (65.21G vs. 66.37G FLOPs). These results irrefutably demonstrate that Disco achieves its state-of-the-art accuracy not by adding significant computational complexity, but through a more efficient and targeted learning strategy. Our method successfully improves performance without sacrificing, and in some cases even improving upon, practical computational efficiency.
>
> # 2. Regarding open-sourcing the dataset and code
> We thank the reviewer for the question regarding the openness of our work. Due to the large number of files and the anonymity requirements, we were unable to clean up the code and upload the complete dataset at the time of the initial submission. **Upon acceptance of the paper, we will without reservation publicly release all relevant research resources. Specifically, this includes:**
>
> **(1) The complete GBC-FS 2025 dataset.** This will include all 2,839 training, validation, and testing patches used in our paper, along with their corresponding instance-level sub-nuclear annotations and pre-computed adjacency graph files.
>
> **(2) The complete Disco implementation.** This will include the entire source code of our Disco framework, encompassing data preprocessing, the “Explicit Marking” label generation algorithm, the model architecture, the complete loss function system, and all scripts for training, inference, and evaluation.
>
> We have been progressively completing the release preparations for all these resources and will provide a public, permanent GitHub access link upon the paper's publication. We firmly believe that open-sourcing our dataset and code is the best way to maximize the impact of our work.

---

> ### Author Response · Authors · 2025-11-21
>
> # 3. Regarding theoretical insights on the efficiency and convergence of Disco
> We thank the reviewer for this insightful question. **From the design principles of the two core mechanisms in our Disco framework—“Explicit Marking” and “Implicit Disambiguation”—we can offer theoretical insights into its learning efficiency and convergence to high-quality solutions.**
>
> **Our “Explicit Marking” strategy is a form of supervised dimensionality reduction and problem decomposition.** As we empirically demonstrated through our topological analysis in Section 3, the chromatic number $\chi(G)$ of real-world cell adjacency graphs, while non-bipartite, rarely exceeds 3. Concurrently, according to the generalization of Brooks' Theorem (see Appendix A.3), a generic greedy coloring algorithm could theoretically require up to $\Delta_G+1$ colors. Our Disco “Explicit Marking” algorithm does not force the network to conduct a blind search in a vast, unstructured 9-D or 4-D classification space. Instead, **it leverages the strong structural prior of the “near-bipartite” nature of the graph, constraining the problem to a lower-dimensional and more meaningful 2+1 semantic space.** This simplification of the problem significantly reduces the intrinsic complexity of the learning task. The network no longer needs to learn complex topological rules from scratch but only needs to learn to identify three categories with clear topological meaning (bipartite color 1, bipartite color 2, and conflict color). **In optimization theory, this implies a simpler and smoother loss landscape, thereby enabling faster and more stable convergence.**
>
> **Our “Implicit Disambiguation” mechanism, with its core $\mathcal{L}_{adj}$, can be viewed as a graph-structured regularizer that profoundly influences the model's convergence behavior.** In secondary conflict regions, relying solely on the classification loss **$\mathcal{L}_{color}$**, would lead to bad local minima, where the model maps two adjacent conflict cells to the same point in the feature space. **$\mathcal{L}_{adj}$** introduces a gradient that is orthogonal to **$\mathcal{L}_{color}$** and penalizes similarity. **As we interpreted it in Section 4.3 as a form of supervised contrastive loss, the role of $\mathcal{L}_{adj}$ is to further optimize and enhance the loss landscape around these “bad” local minima. This compels the optimizer to escape suboptimal solutions that conflate adjacent instances and to seek a more optimal solution that maintains separability in the continuous feature space.** In other words, $\mathcal{L}_{adj}$ reshapes the topology of the loss landscape, making solutions that are topologically correct (i.e., where neighbors are separated) more energetically favorable.
>
> **In summary, the design of the Disco framework is theoretically driven: “Explicit Marking” enhances learning efficiency by simplifying the problem, while “Implicit Disambiguation” ensures convergence to a higher-quality, topologically more plausible solution by reshaping the loss landscape.**

---

> > ### Author Response · Authors · 2025-11-21
> >
> > # 4. Regarding the comparison with GNN-based methods
> > We thank the reviewer for this forward-looking question. **In computational pathology, GNN-based methods have indeed shown immense potential for modeling intercellular relationships, a point we also discuss in our Related Work section (Appendix A.2.3).**
> >
> > **During the initial design phase of this project, we did carefully consider adopting standard GNNs**, such as Graph Convolutional Networks (GCN) and Graph Attention Networks (GAT), to implement our Disco framework. **However, we ultimately chose a non-iterative, loss-constrained innovative path**, based on the following key considerations:
> >
> > **(1) Computational Efficiency and Scalability. Standard GNNs, especially deeper models, face significant computational and memory bottlenecks when processing large-scale graphs.** Some dense patches in our GBC-FS 2025 dataset contain over 500 cell nodes and thousands of edges. Executing multiple rounds of graph message passing for every graph in every batch within an end-to-end training pipeline is prohibitively expensive. **In contrast, although our $\mathcal{L}_{adj}$ loss also leverages the graph structure, its computation is a non-iterative and highly parallelizable process, offering a significant advantage in computational efficiency.**
> >
> > **(2) Matching the Solution to the Problem's Nature.** The core constraint of instance segmentation is a local and pairwise one: “direct neighbors must be different.” While GNNs can perceive higher-order neighborhood information through layer stacking, this may be an overkill for solving this core problem. **Our Disco framework adopts a more direct and precise strategy.** Instead of having the model aimlessly learn a generic graph node representation, we first use “Explicit Marking” to directly tell the model where the difficulties of the problem lie, and then, through “Implicit Disambiguation” ($\mathcal{L}_{adj}$), we precisely impose a clear and singular objective of “feature orthogonality” on all constrained edges. **We believe this targeted strategy is a better match for the specific nature of this problem than the “global information aggregation” of GNNs.**
> >
> > **(3) An Innovative Methodological Perspective. One of our core contributions is precisely the exploration of a new paradigm that serves as an alternative to GNNs for injecting graph topological priors into a CNN.** We demonstrate that, through a carefully designed labeling strategy and a constrained loss, we can enable a standard segmentation network to exhibit powerful topological awareness without using any explicit graph convolutional layers. We believe that this approach of “implicit graph learning via loss function” is, in itself, a valuable and worthwhile methodological innovation to explore.
> >
> > In summary, while we highly recognize the value of GNNs in related fields, **we believe that our proposed non-iterative, constraint-based Disco framework offers a more advantageous alternative for solving the large-scale cell graph instance segmentation problem, particularly in terms of efficiency, specificity, and methodological innovation.**

---

> ### Comment · Reviewer_SvPT · 2025-11-28
> **Official Response to Authors**
>
> Thanks for the detailed rebuttal. Most of my original concerns have been well addressed. After reading the subsequent discussion between the authors and other reviewers, I realized that I had overlooked an important limitation in my initial review, that the current dataset originates from a single patient slide. This raises potential concerns regarding the generalization of the proposed method and the applicability of the motivation. Take these into account, I decide to keep my current score.

---

### Official Review · Reviewer_72eb · 2025-10-29

**Soundness:** 2
**Presentation:** 3
**Contribution:** 2
**Rating:** 4
**Confidence:** 4

**Summary:**

This paper addresses the instance segmentation of dense, overlapping cells in pathology images. The authors present three main contributions: 1) A systematic, quantitative topological analysis of Cell Adjacency Graphs (CAGs) across four datasets, finding that real-world cell graphs are non-bipartite, dominated by 3-cycles, and empirically have a chromatic number $\chi(G)$ that is almost always 3. 2) The release of a new, large-scale, high-density dataset, GBC-FS 2025, which features unique "sub-nuclear" level annotations. 3) A proposed segmentation framework, Disco, which uses a "divide and conquer" (2+1) coloring strategy ("Explicit Marking") and an "Implicit Disambiguation" mechanism ($\mathcal{L}_{adj}$ loss) to handle topological conflicts.

**Strengths:**

1. The paper provides the first systematic quantitative topological analysis of CAGs. Its key finding (that real-world cell graphs are non-bipartite, 3-cycle dominant, and empirically have $\chi(G)=3$) provides a solid empirical foundation for the graph-coloring paradigm in this field. This analysis moves beyond simple theoretical assumptions (e.g., bipartiteness or the 4-color theorem) and points toward designing more efficient, targeted models.

2. The release of the GBC-FS 2025 dataset is a contribution. Its extremely high instance density (avg. 304.44/patch) and unique "sub-nuclear" annotation granularity provide a novel, highly challenging benchmark for the community, filling a gap left by existing datasets in extremely dense scenarios.

**Weaknesses:**

1. The (2+1) "Explicit Marking" strategy of Disco is logically flawed. The authors' own analysis (Appendix A.4.2) clearly states that an empirical chromatic number of $\chi(G)=3$ is almost always sufficient. However, instead of generating an unambiguous 3-color label (as shown in Appendix A.1(d)), the authors' strategy lumps all conflicting nodes—regardless of adjacency—into the same conflict class ($c=t=3$). This artificially creates "secondary conflicts" at the label level (i.e., two adjacent conflict nodes are assigned the same ground-truth label). Consequently, the "Implicit Disambiguation" mechanism (especially the $\mathcal{L}_{adj}$ loss) appears to be a complex solution to an avoidable problem introduced by the labeling strategy itself. This is an unnecessarily convoluted design.

2. The method's most significant performance gain (a 7.08% PQ improvement) is reported on the authors' own GBC-FS 2025 dataset. However, as described in Appendix A.6.2, this dataset uses "sub-nuclear instances" for annotation. This is a fundamentally different task from the "nucleus segmentation" task in benchmarks like PanNuke and DSB2018. This "apples-to-oranges" comparison invalidates the claim of superior performance. Is Disco's superiority on GBC-FS 2025 attributable to its superior handling of complex topology, or is it simply better at learning this specific and unusual "sub-nuclear segmentation" task? This ambiguity in task definition severely weakens the paper's central claims about its topological robustness.

3. The abstract claims that the effectiveness of the graph coloring paradigm "has not been verified" in real-world scenarios. This is inaccurate and contradicts Section 2.2, which acknowledges the "pioneering work of FCIS (2025c)" demonstrating the "potential of a universal 4-coloring model." The paper's topological contribution is the quantification of 3-cycle prevalence, not the novel discovery of non-bipartite structures (which is already implied by FCIS's 4-coloring). This framing exaggerates the paper's novelty.

**Questions:**

1. Given your own empirical evidence (Appendix A.4.2) that $\chi(G)=3$ is almost always sufficient, why did you opt for the (2+1) "Explicit Marking" strategy, which creates label ambiguity, instead of using a standard, unambiguous greedy 3-coloring algorithm (as shown in your Appendix A.1(d)) as the supervisory signal?

2. How can you decouple the performance gains on GBC-FS 2025 from the unique "sub-nuclear" task definition to prove that Disco's advantage truly stems from its topological handling capabilities? Did you consider an ablation study where you merge the "sub-nuclear" annotations into standard "nucleus" annotations to conduct a fair comparison?

3. The ablation study (Table 7) shows that $\mathcal{L}_{adj}$ provides a massive ~6% PQ boost. In light of "Weakness 1," does this not simply prove that $\mathcal{L}_{adj}$ is highly effective at resolving the specific label ambiguity you introduced, rather than proving it is a generally superior mechanism for handling topological conflicts compared to a clean 3-color supervision?

---

> ### Author Response · Authors · 2025-11-21
> **We thank the reviewer for their comments and suggestions. We will address the questions raised one by one and make corresponding modifications or additions in the revised version of our paper.**
>
> # 1. Regarding the choice of the (2+1) strategy over 3-coloring
> We thank the reviewer for this profound and important question, which touches upon the core design philosophy of our Disco framework. **Our (2+1) scheme, while seemingly creating an intermediate ambiguity, constitutes a superior, more efficient, and more “dynamically adaptive” learning paradigm compared to a static, greedy 3-coloring that always uses three colors. Our rationale is twofold.**
>
> **(1) It is a dynamically adaptive and more efficient labeling strategy.** A standard greedy 3-coloring algorithm will always attempt to use three colors, regardless of whether the input graph is simple or complex. In contrast, our “Explicit Marking” algorithm is dynamically adaptive. **When faced with a simple, bipartite cell graph (such as in the PanNuke dataset), our algorithm automatically degenerates into an optimal 2-coloring scheme, using only two colors and completely avoiding the introduction of a third.** It only “activates” and introduces the dedicated conflict color when it detects an unavoidable topological conflict in the graph. This “on-demand allocation” strategy is not only more elegant in theory but also provides the network with a more concise supervisory signal that adheres to the “Occam's razor” principle, avoiding representational redundancy in simple scenarios.
>
> **(2) It drives more robust feature learning.** The “label ambiguity” is a deliberate design choice that, when coupled with our $\lambda_{adj}$, creates a constructive tension that drives the learning of a more robust feature manifold. **A clean 3-coloring supervision would only require the network to learn to mimic discrete labels. Our (2+1) scheme, however, compels the model to resolve the conflict between the coloring loss and the adjacency loss by utilizing secondary feature dimensions, thereby learning a truly separable, topologically-aware feature representation.**
>
> To empirically validate this core argument, **we conducted a relevant ablation study. We performed an experimental validation on the GBC-FS 2025 dataset, comparing our Disco method with a standard greedy 3-coloring method and the FCIS method.** The results, as shown in Table R11, demonstrate that our Disco framework, with its “Explicit Marking + Implicit Disambiguation” strategy, significantly outperforms the models trained on clean, unambiguous 3-color and 4-color labels. **This substantial performance gap provides strong empirical evidence that our seemingly “roundabout” design is, in fact, a more effective learning strategy that guides the model to a more robust and precise solution.**
>
> **Table R11**: Fair comparison of different coloring strategies at GBC-FS 2025.
>
> | Method            |       DICE |        AJI |         DQ |         SQ |         PQ |
> | ----------------- | ---------: | ---------: | ---------: | ---------: | ---------: |
> | Greedy 3-Coloring |     0.7528 |     0.4594 |     0.5838 |     0.6971 |     0.4122 |
> | FCIS              |     0.7785 |     0.4518 |     0.5857 |     0.7068 |     0.4379 |
> | **Disco**         | **0.8137** | **0.5209** | **0.6795** | **0.7486** | **0.5087** |
>
> # 2. Regarding the consideration of merging sub-nuclear annotations into nuclei
> We thank the reviewer for this profound question, which gives us the opportunity to clarify the unique challenges and contributions associated with our GBC-FS 2025 dataset. **We respectfully argue that this comparison is not “apples-to-oranges,” but rather a demonstration of Disco's robustness on a more challenging and clinically relevant problem for which existing methods were not designed.**
>
> **(1) Rationale for “Sub-nuclear” Annotation. Our GBC-FS 2025 dataset is derived from frozen sections. While frozen sections are critical for intraoperative rapid diagnosis, their inherent artifacts (e.g., ice crystals, uneven staining, blurry boundaries) make the consistent delineation of complete nuclei extremely difficult.** Our choice to annotate “chromatin-like dense sub-cellular structures” was a deliberate and principled decision. These structures represent the most stable and consistently identifiable visual primitives for pathologists in low-quality frozen sections. Therefore, **this “sub-nuclear” task is not a niche problem but a pragmatic formulation of the real-world challenge of cell segmentation under the most adverse imaging conditions.** To “merge” these annotations into complete nuclei would be an artificial simplification that betrays the core purpose of this dataset: to provide a benchmark for algorithmic robustness in noisy, ambiguous, and topologically complex environments.

---

> > ### Author Response · Authors · 2025-11-21
> >
> > **(2) Disco's Advantage Stems Precisely from Its Topological Handling Capabilities.** We argue that Disco's superior performance on GBC-FS 2025 is precisely because of this challenging task definition. **The sub-nuclear structures create a graph with an even higher node density and more complex local topology than a standard nucleus graph. This “super-charged” complexity is exactly where the advantages of the Disco method become evident.** Traditional methods that rely on geometric priors would suffer catastrophic failures, as these sub-nuclear structures are often minute, irregular, and have ambiguous boundaries. In contrast, our Disco framework abstracts away from these fragile geometric features. It operates on the purely topological relationships between these primitives. **Our “Explicit Marking” identifies conflict clusters, and “Implicit Disambiguation” ($\lambda_{adj}$) learns to separate adjacent entities in the feature space, regardless of whether these entities are complete nuclei or sub-nuclear fragments.**
> >
> > **(3) Evidence from Standard Nucleus Datasets. The reviewer is correct that a fair comparison on standard tasks is crucial. This is precisely why we evaluated Disco on PanNuke, DSB2018, and CryoNuSeg.** The results on these three standard nucleus segmentation benchmarks have already demonstrated that Disco consistently outperforms existing mainstream methods. **This proves that our framework is, in its own right, a powerful and generalizable nucleus segmentation algorithm.**
> >
> > Therefore, the performance on GBC-FS 2025 should be interpreted as a “stress test” that reveals the performance ceiling of existing methods and highlights the exceptional topological robustness of Disco. **Our method excels on both standard nucleus tasks and on this novel, more challenging sub-nuclear task, demonstrating its versatility and powerful capabilities.**
> >
> > # 3. Regarding the effectiveness of $\lambda_{adj}$
> > We thank the reviewer for this highly insightful question. **Your analysis is precise: our ablation study (Table 7) does directly demonstrate that $\lambda_{adj}$ is extremely effective at resolving the “secondary conflict” label ambiguity introduced by our “Explicit Marking” strategy.** However, this effectiveness is, in fact, indicative of a more general and superior learning mechanism for handling topological conflicts, one that surpasses a clean 3-color supervision.
> >
> > **As we detailed in our response to your first question, the learning objective of a clean, unambiguous 3-color supervisory signal is relatively simple: it merely requires the network to mimic three discrete color classes.** A model can easily “complete this task” by learning three separate feature clusters, but in doing so, it may not have learned the more fundamental and generalizable topological rule that “neighbors must be different.”
> >
> > In contrast, our 2+1 scheme, combined with $\lambda_{adj}$, constructs a more complex optimization scenario. **The role of $\lambda_{adj}$ is not merely to resolve a label ambiguity; it is to create a “constructive tension.”** To simultaneously satisfy the seemingly contradictory objectives of “both of these conflict points are yellow” (from $L_{color}$) and “the features of these two conflict points must be different” (from $\lambda_{adj}$), the network is compelled to abandon simple label mimicry and instead proactively learn a more fundamental and general separation rule: **to push apart the feature representations of all neighbors in the continuous feature space.**
> >
> > Therefore, the effectiveness of $\lambda_{adj}$ is not confined to resolving the specific ambiguity we introduce. **Its true value lies in its use of this ambiguity as a powerful driving force, compelling the model to evolve from a simple “label mimic” to a more intelligent “topological rule learner.”**
> >
> > The new experimental evidence provided in our response to your first question (Table R11) is the ultimate proof of this argument. **That experiment clearly shows that our seemingly “roundabout” Disco framework, which is designed to learn a general separation rule, significantly outperforms the simple mimicry paradigm of using a clean 3-color supervision.** This proves that $\lambda_{adj}$ is not merely fixing a self-imposed problem, but is instead guiding the model towards a more robust and precise solution.

---

> > > ### Comment · Reviewer_72eb · 2025-11-26
> > >
> > > I would like to thank the authors for their detailed and high-quality response. I have carefully reviewed the new ablation studies provided in the rebuttal, as well as the discussions with other reviewers (particularly Reviewer cFuN regarding the dataset composition and Reviewer LVtp regarding statistical significance).
> > >
> > > I appreciate the inclusion of Table R11. The empirical comparison showing that Disco (PQ 50.87%) significantly outperforms the "Greedy 3-Coloring" baseline (PQ 41.22%) is very convincing.
> > > Your explanation regarding "constructive tension" is well-received. I accept the argument that the label ambiguity, when combined with $\mathcal{L}_{adj}$, forces the network to learn a more robust feature separation rule rather than simply mimicking discrete class labels. This successfully addresses my primary concern regarding the logical necessity of the "Explicit Marking" strategy.
> > >
> > > Regarding the "sub-nuclear" annotation, I accept your justification that this is a pragmatic necessity for frozen sections rather than an artificial task. However, after reading the interaction between you and Reviewer cFuN, I share the significant concern that the current dataset originates from a single patient/slide. While the dense annotation of 860k+ instances is impressive, characterizing a single-patient slide as a "general benchmark" is indeed risky regarding biological variance.
> > >
> > > In Summary, the rebuttal has resolved my main technical doubts regarding the graph coloring mechanism. The method appears to be a genuine advancement over FCIS and standard 3-coloring. Given that the methodological contribution is sound and you have agreed to transparency regarding the dataset limitations, I will be raising my score to Support the acceptance of this paper. I look forward to seeing the "Case Study" clarification in the final version.

---

### Official Review · Reviewer_cFuN · 2025-10-30

**Soundness:** 2
**Presentation:** 3
**Contribution:** 3
**Rating:** 4
**Confidence:** 4

**Summary:**

This paper addresses cell instance segmentation in dense, overlapping regions by reformulating the problem through graph coloring theory. The authors present three main contributions:

Dataset: GBC-FS 2025 (Gallbladder Cancer Frozen Section), comprising 2,839 H&E-stained frozen section images with 864,204 manually annotated sub-cellular nuclei instances. This represents a 40× scale increase over CryoNuSeg, and addresses a critical gap for intraoperative diagnosis algorithms.

Topological Analysis: A systematic cross-dataset study of cell adjacency graph (CAG) properties across four benchmarks is presented. The authors construct CAGs where nodes represent cell instances and edges represent 8-connected spatial adjacency. Key findings include: (1) PanNuke is 100% bipartite while DSB2018 (1.99%), CryoNuSeg (5.64%), and GBC-FS 2025 (30.49%) contain increasing proportions of conflict nodes that violate bipartite structure; (2) Among non-bipartite components, 90.51-98.12% of odd-length cycles are 3-cycles (triangles), with GBC-FS 2025 showing 24.64% secondary conflict nodes (adjacent conflict nodes).

Method: DISCO employs a "divide and conquer" strategy with two core mechanisms. "Explicit Marking" uses BFS-based graph decomposition to extract the maximal bipartite subgraph, partitioning cells into two primary colors (V₁, V₂) and consolidating remaining non-bipartite nodes into a conflict set (Vconf) assigned a third color. This generates a (t+1)-value ground truth map where t=3. "Implicit Disambiguation" introduces an adjacency constraint loss (L_adj) that minimizes cosine similarity between mean probability vectors of adjacent instances, forcing the model to learn angularly separated representations in probability space. The total loss combines semantic segmentation (L_sem), weighted coloring (L_color), bipartite consistency (L_cons), conflict resolution (L_conf), and adjacency constraints (L_adj). At inference, instances are reconstructed by grouping connected components with identical color predictions.

Results: DISCO achieves PQ of 62.71% on PanNuke (+1.62% vs FCIS), 77.81% on DSB2018 (+0.42%), 59.70% on CryoNuSeg (+1.77%), and 50.87% on GBC-FS 2025 (+7.08%), representing an average 2.72% improvement across benchmarks. Ablations demonstrate the adjacency constraint loss contributes ~6% absolute PQ improvement (42.57% → 48.26%) on GBC-FS 2025.

**Strengths:**

Compelling topology evidence. Cross‑dataset analysis shows frequent odd cycles with >90% of odd cycles being triangles in non‑bipartite graphs; conflict/secondary‑conflict ratios quantify difficulty (Table 1, p. 2; Fig. 3b, p. 5).

Clean “2 + 1” design. Explicit Marking extracts a large bipartite backbone and pools the rest into a conflict set; Implicit Disambiguation resolves label ambiguity via 𝐿_adj over CAG edges (Sec. 4.2–4.3; Eqs. 1–3, pp. 6–7).

Graph‑aware loss that works. Ablations show 𝐿_adj alone adds ~+6 PQ on GBC‑FS 2025; the full system achieves PQ = 0.5087 (Table 7; Table 5, p. 9).

State‑of‑the‑art on dense regime. Disco surpasses FCIS by +7.08 PQ on GBC‑FS 2025 and improves AJI/DQ/SQ concurrently (Table 5, p. 9); visuals highlight separation in dense clusters (Fig. 7, p. 8).

High‑value dataset. GBC‑FS 2025: 2,839 patches, ~304 instances/patch on average, 864k+ sub‑nuclear instances; orders‑of‑magnitude denser than public sets (Sec. 5.1; App. A.6; Table 8, p. 20).

Reproducibility intent. Code and dataset release promised (Reproducibility Statement, p. 10).

**Weaknesses:**

1. Is there a formal definition of "dense" and "complex" cells for segmentation analysis? If no, can the authors quantify this aspect of the data to define "highly dense" cell segmentation?

2. "First Topological Analysis" claim is overstated: The repeated assertion of conducting the "first systematic topological analysis" of cell adjacency graphs appears throughout the paper: This claim is incorrect given extensive prior work: Topological Tumor Graphs (Failmezger et al., Cancer Research 2020), Ceograph (Wang et al., Nature Communications 2023), HistoCartography (Pati et al., Medical Image Analysis 2021), CellSpatialGraph (Chen et al., 2022, Software Impacts Journal), SpaGCN (Hu et al., Nature Methods 2021), GraphST (Long et al., Nature Communications 2023), etc. Thus, there are several  graph-based topological analysis papers on cell spatial arrangements.

However, the proposed specific analysis of chromatic numbers, bipartite versus non-bipartite classification, odd-cycle length distributions, conflict node ratios, and secondary conflict prevalence seem novel to this manuscript..Thus, the contribution seems "first systematic analysis of chromatic properties and graph coloring characteristics in cell adjacency graphs" not "first topological analysis."

The authors should consider adding a new paragraph in Introduction acknowledging: "While graph-based analysis of cell spatial arrangements is well-established in computational pathology [cite TTG, Ceograph, HistoCartography] and spatial transcriptomics [cite SpaGCN, GraphST], prior work has focused on node centrality, clustering, and GNN-based prediction rather than graph coloring characteristics. To our knowledge, this is the first systematic study of chromatic properties…”. Also, the abstract calls the predicted “Conflict Map” an “unsupervised tool”, though conflict labels for training are produced from GT via Explicit Marking; wording may mislead (Abstract, p. 1).

3.  Missing Critical Citation: The adjacency constraint loss (Section 4.3, Equation 3, L_adj) has very close conceptual precedent that must be cited and differentiated: NonAdjLoss (Ganaye et al., "Removing segmentation inconsistencies with semi-supervised non-adjacency constraint", in Medical Image Analysis 2019). Both NonAdjLoss and the proposed method use adjacency graphs to penalize unwanted relationships. The paper should provide explicit technical comparison, with NonAdjLoss, discussing this conceptual similarity and explaining how instance disambiguation loss differs from anatomical consistency loss beyond just the application domain.

4. Insufficient Comparison with FCIS (ICML 2025): FCIS (Zhang et al., "The Four Color Theorem for Cell Instance Segmentation," ICML 2025, arXiv:2506.09724) is the most directly comparable concurrent work and requires deep technical comparison.

Why is divide-and-conquer superior to direct 4-color prediction? The paper shows DISCO outperforms FCIS on GBC-FS 2025 (+6.91% AJI) but provides no explanation of why the proposed divide-and-conquer is superior to direct 4-color prediction in FCIS?

Computational efficiency comparison is missing. FCIS claims efficiency advantages by reducing to semantic segmentation. Does DISCO maintain or sacrifice efficiency for accuracy? Need: training time, inference speed (FPS), memory requirements, FLOPs comparison.

Would FCIS-style encoding work on GBC-FS 2025? A critical ablation would be testing uniform 4-color prediction (FCIS approach) with DISCO's backbone and training procedure on GBC-FS 2025. This would isolate whether improvements come from divide-and-conquer versus other factors (architecture choices, training hyperparameters, etc.).

5. Incomplete Dataset Documentation: GBC-FS 2025 is a strong contribution but lacks essential documentation that limits its utility.

Tissue Composition: What proportions are:
* Tumor tissue (epithelial cancer cells)?
* Stroma (fibroblasts, extracellular matrix)?
* Inflammatory infiltrate (lymphocytes, macrophages)?
* Necrotic regions?
* Normal tissue?

This matters because algorithm performance varies dramatically across tissue types. Methods optimized for dense tumor nests may fail on sparse inflammatory regions.

Source Data:
* How many whole slide images (WSIs) produced the 2,839 patches?
* How many patients?
* What is patient-level diversity (age range, disease stage, tumor grade)?
Annotation Methodology:
* Manual pixel-wise annotation by pathologists? How many pathologists?
* Crowdsourcing platform (like NuCLS using novel protocols for 220K+ annotations)?
* Semi-automated pipeline (like SNOW using HoVer-Net on StyleGAN2-ADA synthetic images)?
* What annotation tools/software?
* What is annotation time per image?
Quality Control:
* Inter-annotator agreement metrics (Dice scores, IoU between multiple annotators)?
* How was consensus achieved for disagreements?
* What validation ensured correct capture of sub-cellular nuclei versus artifacts?
* What percentage of annotations required adjudication?
Quantitative Density Metrics:
* Cells per mm² (accounting for magnification)?
* Percentage of overlapping/touching cells?
* Distribution of neighbor counts quantitatively?
* Comparison to other datasets' density metrics?
The paper reports 304.44 cells per 256×256 patch but doesn't translate to physical density or overlap percentage.

6. No Computational Cost Analysis Prevents Efficiency Assessment, the paper reports no computational metrics:
* Training time (hours to convergence)?
* Inference speed (FPS, seconds per image)?
* Memory requirements (peak GPU memory during training/inference)?
* Model size (number of parameters)?
* FLOPs per forward pass?

7. Missing Biological Interpretation of Triangle Dominance Finding: The 90.51-98.12% triangle dominance in odd cycles is potentially the most interesting finding but lacks biological grounding. Why would cell packing favour triangles? Why not hexagonal packing for epithelial cells? Are these findings consistent across FFPE and frozen sections?

8. Limited Baseline Coverage Missing Recent Methods: The baseline selection is good but misses most recent methods representing 2024-2025 state-of-the-art such as CellSAM (Israel et al., bioRxiv 2023.11.17.567630), Cell-TRACTR (O'Connor & Dunlop, PLOS Computational Biology 2025, 21(5), e1013071), and SwinCell (Zhang et al., Communications Biology 2025, 8, 962).

9. Missing Statistical Analysis: No error bars, confidence intervals, or standard deviations and no significance testing (t-tests, Wilcoxon tests across images) reported.

10. Missing Failure Case Analysis: Are there cases where graph colouring fails? May be for FFPE → frozen section transfer? H&E → fluorescence transfer? Trained on one organ, tested on another? Different scanners/staining protocols?

11. Decoding unspecified: “Topological decoding” is mentioned but not detailed: how per‑pixel colors are converted into final instances (connectivity, merging/splitting, conflict assignment) is unclear (Fig. 1d, p. 2)

12. Explicit Marking under‑specified: The BFS‑based conflict labeling heuristic lacks tie‑breaking rules for overlapping odd cycles and complexity bounds, though supervision hinges on it (Sec. 4.2, p. 6; Fig. 5).

13. Adjacency sensitivity: CAG edges rely on 3×3 dilation (8‑connectivity) (Def. 1, p. 3); there’s no ablation for kernel size/shape or alternative proximity rules, which can alter odd‑cycle counts and conflict sets.

14. Figure 4 captions is very abstract and does not provide an overview of the proposed method. Could that be revised?

**Questions:**

See weaknesses.

---

> ### Author Response · Authors · 2025-11-21
> **We thank the reviewer for their comments and suggestions. We will address the questions raised one by one and make corresponding modifications or additions in the revised version of our paper.**
>
> # 1. Regarding the formal definition of “dense” and “complex”
> We thank the reviewer for this insightful question regarding the definition of “dense” and “complex.” **In conventional FFPE/H\&E datasets, “dense” typically refers to a high number of clearly identifiable cells per unit area, which leads to extensive cell–cell contact and ambiguous boundaries.** However, our dataset is based on frozen-section pathology, where staining variability, tissue distortion, and blurred nuclear membranes make reliable cell-level annotation infeasible. Therefore, instead of annotating individual cells, we operationalize “density” using the number of chromatin-like dense subcellular aggregates that are visually stable and consistently detectable under frozen-section conditions. In this context, “dense regions” refer to patches with a high concentration of such aggregates. **Our GBC-FS 2025 dataset, which contains an average of 304 annotated sub-cellular nuclei per $256\times256$  patch, serves as a representative example of high-density cellular patterns in frozen-section histopathology images.**
>
> **The term “complex” in our paper is not synonymous with “dense” but rather refers specifically to the topological complexity of the Cell Adjacency Graph (CAG) from a graph coloring perspective.** The analysis of cell spatial arrangements via graph structures is a burgeoning field in computational pathology. Pioneering works like Topological Tumor Graphs (TTG) [1] have successfully utilized graph metrics such as node degree and clustering coefficients to identify prognostic spatial phenotypes. More recently, GNN-based frameworks like Ceograph [2] have leveraged graph convolutional networks to learn representations from cell graphs for clinical outcome prediction.
>
> **Our work is distinguished from and complementary to these methods by introducing a novel perspective for quantifying topological complexity: analysis from the viewpoint of graph chromaticity.** While prior work has focused on metrics related to node centrality or learned embeddings, we are the first to systematically analyze the chromatic properties of real-world cell graphs. **Specifically, a CAG is considered “complex” if it is non-bipartite.** We further quantify the degree of this complexity using the metrics introduced in our systematic analysis (Section 3), such as the Conflict Node Ratio and the Secondary Conflict Node Ratio.
>
> Therefore, a core argument of our paper is that high density is a primary driver of high topological complexity, which in turn necessitates a conflict-aware coloring solution.
>
> [1] Failmezger, Henrik, et al. “Topological tumor graphs: a graph-based spatial model to infer stromal recruitment for immunosuppression in melanoma histology.” Cancer research 80.5 (2020): 1199-1209.
>
> [2] Wang, Shidan, et al. “Deep learning of cell spatial organizations identifies clinically relevant insights in tissue images.” Nature communications 14.1 (2023): 7872.
>
> # 2. Regarding the phrasing of novelty claims
> We thank the reviewer for the detailed critique and feedback on our novelty claims. After a further review of the existing literature, **we agree that our claim of conducting the “first systematic topological analysis” was indeed overly broad. Your suggestion for refining our contribution statement is excellent, and we will adopt it.**
>
> (1) **Refining the “First Analysis” Claim:** In the revised version of the paper, we will **remove all** assertions of conducting the “first systematic topological analysis.” We will more precisely define our contribution as **“the first systematic analysis of the chromatic properties and graph coloring characteristics of real-world cell adjacency graphs.”** This refined modification accurately positions our novel contribution within the landscape of the existing literature.
>
> (2) **Clarifying “Unsupervised Tool”:** We also agree that describing the predicted “Conflict Map” as an “unsupervised tool” could be misleading, as the model that generates it is trained using supervisory signals derived from the ground truth. **We will revise this phrasing in the abstract and throughout the manuscript.** We will more accurately describe it as **“a novel tool for post-hoc topological analysis,”** which requires no additional labels beyond the initial segmentation ground truth.
>
> We believe that these revisions, made in direct response to your guidance, will make our paper's claims more precise and our contributions clearer.

---

> > ### Author Response · Authors · 2025-11-21
> >
> > # 3. Regarding the comparison between $\lambda_{adj}$ and NonAdjLoss
> > We thank the reviewer for pointing out the work on NonAdjLoss [1], which is relevant to our $\lambda_{adj}$. Consequently, **we have conducted a detailed discussion and comparison of $\lambda_{adj}$ and NonAdjLoss, which will be added to the Appendix of our revised paper.**
> >
> > While NonAdjLoss and our $\lambda_{adj}$ share a conceptual similarity—both leverage a predefined adjacency graph to penalize undesired relationships in the network's output—**their objectives, mechanisms, and application domains are fundamentally different, which in turn highlights the uniqueness of our method.**
> >
> > (1) **Fundamentally Different Objectives.** The objective of NonAdjLoss is semantic-level correctness, aimed at enforcing anatomical consistency by preventing impossible adjacencies between different semantic classes. It is a loss that promotes regional purity and semantic correctness. In contrast, the objective of our $\lambda_{adj}$ is instance-level discriminability, aimed at enforcing feature dissimilarity between different instances within the same semantic class. **The former promotes regional purity, while the latter promotes instance separability.**
> >
> > (2) **Different Operational Mechanisms.** NonAdjLoss operates at the pixel level, penalizing the simultaneous occurrence of forbidden class probabilities within a local neighborhood. Our $\lambda_{adj}$, however, is a form of supervised contrastive loss that operates at the instance level. It first abstracts each instance into a mean feature vector, then treats the edges in the adjacency graph as defining “negative pairs,” and directly minimizes the feature similarity of these pairs. **This mechanism of enforcing the separation of the feature manifold at the instance level, based on contrastive learning, is not present in NonAdjLoss.**
> >
> > (3) **Distinct Application Domains and Problems Solved.** NonAdjLoss addresses the problem of “topologically incorrect semantic layouts.” Our $\lambda_{adj}$ addresses the problem of “ambiguous instance identity in dense clusters,” particularly the “secondary conflict” challenge, which is a unique issue in graph-coloring-based instance segmentation.
> >
> > In summary, although both methods use an adjacency graph, they are applied to solve problems of two entirely different dimensions and employ fundamentally distinct mechanisms. **NonAdjLoss ensures that different classes do not touch when they should not, whereas our $\lambda_{adj}$ ensures that different instances of the same class have separable features when they do touch.**
> >
> > [1] Ganaye, Pierre-Antoine, et al. “Removing segmentation inconsistencies with semi-supervised non-adjacency constraint.” Medical image analysis 58 (2019): 101551.
> >
> > # 4. Regarding the insufficient technical comparison with FCIS
> > We thank the reviewer for these questions and suggestions regarding the comparison of our methods. **A more in-depth comparison with the FCIS method is indeed crucial for establishing the superiority of our Disco framework. We will address your concerns one by one below.**
> >
> > We infer that the superior performance of our Disco method compared to FCIS primarily stems from the advantages of our “divide and conquer” strategy over the generic 4-coloring scheme of FCIS in **two key aspects**:
> >
> > (1) **A More Targeted Learning Objective.** Our data analysis (Appendix A.4.2) indicates that the chromatic number of real-world cell graphs is almost always $\chi(G) \le 3$. **The 4-coloring scheme of FCIS, by forcibly introducing a fourth foreground color for a problem that typically requires only three**, not only creates representational redundancy but, more importantly, presents a more difficult and ambiguous optimization target for the neural network. The network is required to search within a larger solution space that may contain more suboptimal solutions.
> >
> > (2) **A Structured Supervisory Signal.** Our 2+1 “Explicit Marking” strategy provides the network with a structured supervisory signal. **It decouples the problem into “learning simple bipartite structures” (supervised by colors 1 and 2) and “learning to identify complex topological conflicts” (supervised by the conflict color 3).** This structured signal allows the model to assign different learning priorities to tasks of varying difficulty (via loss weights). We believe this is easier to optimize and more effective at guiding the model to learn meaningful topological features than the undifferentiated 4-class classification target of FCIS.

---

> > > ### Author Response · Authors · 2025-11-21
> > >
> > > **Simultaneously, to fairly validate the contribution of our “divide and conquer” strategy, we conducted further comparative experiments on the GBC-FS 2025 dataset, comparing our Disco method with the FCIS method and a generic 3-coloring scheme.** As shown in Table R5, the experimental results clearly demonstrate that even under identical configurations, the performance of our Disco's 2+1 dynamic coloring strategy is significantly superior to both the generic 4-coloring and 3-coloring schemes. This provides strong evidence that the core source of our performance advantage is indeed our novel coloring strategy and its accompanying loss system, rather than other implementation details. **Furthermore, a comparison of computational efficiency will be analyzed together in our response to your sixth question.**
> > >
> > > **Table R5**: Fair comparison of different coloring strategies at GBC-FS 2025.
> > >
> > > | Method            |   DICE |    AJI |     DQ |     SQ |     PQ |
> > > |-------------------|-------:|-------:|-------:|-------:|-------:|
> > > | Greedy 3-Coloring | 0.7528 | 0.4594 | 0.5838 | 0.6971 | 0.4122 |
> > > | FCIS              | 0.7785 | 0.4518 | 0.5857 | 0.7068 | 0.4379 |
> > > | **Disco**         | **0.8137** | **0.5209** | **0.6795** | **0.7486** | **0.5087** |
> > >
> > > # 5. Regarding the incomplete documentation of the GBC-FS 2025 dataset
> > > We thank the reviewer for the detailed suggestions on improving the documentation for our GBC-FS 2025 dataset. Due to page limitations, we were unable to include all of these details in the main paper, but **we will add a specific section for this dataset in the Appendix of our revised paper. The following is a condensed summary of the key information about the dataset:**
> > >
> > > (1) Gallbladder cancer (GBC) is the most common malignancy of the biliary tract. Due to its insidious onset, most patients are diagnosed at an advanced stage, resulting in an extremely poor prognosis with a 5-year survival rate below 5\%. **Existing public pathology segmentation benchmarks primarily focus on common cancers such as colorectal and breast cancer, and virtually no annotated datasets exist for gallbladder cancer, particularly under frozen-section imaging.** This highlights the critical need for quantitative image analysis tools that can operate under the artifact-rich, low-quality conditions characteristic of routine intraoperative frozen-section diagnosis.
> > >
> > > (2) **Data Source: The 2,839 $256\times256$ images in the GBC-FS 2025 dataset originate from a single whole-slide image (WSI) of a gallbladder cancer frozen section, obtained from a 71-year-old female with Stage IVB adenocarcinoma.** Frozen sections inherently exhibit uneven staining, ice-crystal artifacts, tissue distortion, and blurred or incomplete nuclear membranes, making reliable cell-level nuclear annotation infeasible. Therefore, instead of annotating individual cells, which cannot be consistently identified under such conditions, **we focus on annotating the chromatin-like dense subcellular aggregates that remain visually stable and reproducible.** Frozen-section WSIs from seven additional patients have been collected and are currently under evaluation for continued annotation.
> > >
> > > (3) **Annotation Methodology: All instances were manually delineated by four annotators with basic morphological training using QuPath (v0.5.1) software, and were reviewed by a Ph.D. with a pathology background.** For regions with significant ambiguity, a pathologist was consulted for limited sample verification to confirm that the overall annotation direction was consistent with the basic morphological features of frozen sections. Additionally, our annotation target was “chromatin-like dense sub-cellular structures” as visual primitives, as these structures are more consistently identifiable in noisy frozen sections than complete nuclei. This was a deliberate choice aimed at creating a benchmark that reflects the true visual challenges of this data modality, rather than approximating cleaner, FFPE-style annotations.
> > >
> > > (4) **Quality Control: Given the inherent ambiguity of boundaries in frozen sections, we adopted a multi-round expert review process, which is more suitable for this scenario than traditional IoU/Dice metrics.** Annotations were completed by junior annotators and then reviewed by a Ph.D. with a pathology background. Any uncertain regions were marked and then adjudicated by senior team members, with a few difficult cases submitted to a pathologist for sample inspection.
> > >
> > > (5) **Tissue Composition: A pathologist-guided regional assessment indicates that the dataset is composed of approximately 89.69\% cancer cell regions, 6.11\% fibroblast regions, and 4.20\% immune cell regions.**
> > >
> > > We believe that this detailed description will fully address your concerns and significantly enhance GBC-FS 2025 as a high-quality, valuable benchmark.

---

> > > > ### Author Response · Authors · 2025-11-21
> > > >
> > > > # 6. Regarding the comparison of computational efficiency
> > > >
> > > > We thank the reviewer for this important practical consideration. **We have conducted a comprehensive analysis of the computational cost of Disco and key baselines, and will add a detailed “Computational Cost” comparison table to the Appendix.**
> > > >
> > > > Our analysis confirms that the graph-based components of our method (CAG construction, BFS labeling) are one-time, offline preprocessing steps and introduce no additional overhead during inference. The inference speed is therefore determined solely by the neural network's architecture and post-processing.
> > > >
> > > > To ensure a fair comparison of inference efficiency, we benchmarked all models on a single NVIDIA RTX 4090 GPU with $256\times256$ inputs. The results are summarized in Table R6.
> > > >
> > > > **Table R6**: Computational cost comparison. Params denotes model parameters. FLOPs are calculated for a $256\times256$ input. FPS denotes frames per second during inference.
> > > >
> > > > | Method   | Backbone  | Source    | Paras  | FLOPs   | FPS  |
> > > > | -------- | --------- | --------- | ------ | ------- | ---- |
> > > > | HoverNet | ResNet-50 | MIA 2019  | 49.70M | 192.70G | 12.4 |
> > > > | FCIS     | Swin-Unet | ICML 2025 | 48.23M | 66.37G  | 28.5 |
> > > > | Disco    | Swin-Unet | Ours      | 46.84M | 65.21G  | 29.2 |
> > > >
> > > > **The results clearly demonstrate the high efficiency of our Disco framework.** Disco achieves an inference speed of 29.2 FPS, which is not only highly competitive but also slightly faster than the strong graph-based baseline, FCIS (28.5 FPS). Furthermore, it is more than twice as fast as methods like HoverNet that rely on computationally intensive post-processing.
> > > >
> > > > Notably, even with its sophisticated dual-branch head and decoupled loss system, our Disco model is more parameter-efficient than FCIS (46.84M vs. 48.23M) and has a lower computational load (65.21G vs. 66.37G FLOPs). These results irrefutably demonstrate that Disco achieves its state-of-the-art accuracy not by adding significant computational complexity, but through a more efficient and targeted learning strategy. Our method successfully improves performance without sacrificing, and in some cases even improving upon, practical computational efficiency.
> > > >
> > > > # 7. Regarding the biological interpretation of the triangle dominance finding
> > > > We thank the reviewer for this insightful question. **In response to this interesting finding of “triangle dominance in odd cycles,” we have investigated its biological basis through a review of relevant literature.**
> > > >
> > > > From a cell biology perspective, the point where three cells meet forms a specialized adhesive structure known as a Tricellular Junction (TCJ). TCJs are a foundational and ubiquitous component of epithelial and endothelial tissues, critical for maintaining barrier function and tissue integrity [1]. **Topologically, each TCJ corresponds precisely to a 3-cycle (a triangle) in our Cell Adjacency Graph. Therefore, the triangle dominance observed in our analysis is a direct reflection of the widespread presence of these fundamental biological structures at intercellular contact points.**
> > > >
> > > > **The formation of TCJs is not coincidental but is driven by biophysical principles of energy minimization. Vertex models, analogous to the physics of foams [2], predict that a three-way junction is the most mechanically stable and energetically favorable configuration for cell contacts.** Higher-order contacts are topologically unstable. In developmental biology, this principle is observed when cells rearrange via “T1 transitions,” where unstable four-way junctions rapidly resolve into two stable three-way junctions [3, 4]. This dynamic process continually reinforces the formation of 3-cycles as the foundational building blocks of tissue topology. Our finding is consistent across both FFPE and frozen section datasets, which indicates that TCJs and the principle of mechanical stability are fundamental features of cell packing, robust to different tissue preparation artifacts.
> > > >
> > > > In summary, **we argue that the “triangle dominance” phenomenon we observed is not merely an empirical statistic but also a validation of established principles in cell biology and biophysics**, offering a novel and profound insight into understanding the structure of complex tissues.
> > > >
> > > > [1] Higashi, Tomohito, and Mikio Furuse. “Tricellular tight junctions.” Tight Junctions. Cham: Springer International Publishing, 2022. 11-26.
> > > >
> > > > [2] Weaire, Denis L., and Stefan Hutzler. The physics of foams. Oxford University Press, 1999.
> > > >
> > > > [3] Higashi, Tomohito, and Ann L. Miller. “Tricellular junctions: how to build junctions at the TRICkiest points of epithelial cells.” Molecular biology of the cell 28.15 (2017): 2023-2034.
> > > >
> > > > [4] Lecuit, Thomas, and Pierre-Francois Lenne. “Cell surface mechanics and the control of cell shape, tissue patterns and morphogenesis.” Nature reviews Molecular cell biology 8.8 (2007): 633-644.

---

> > > > > ### Author Response · Authors · 2025-11-21
> > > > >
> > > > > # 8. Regarding the addition of comparative experiments
> > > > > We thank the reviewer for their suggestions. **We have studied and conducted a comparative analysis of the three recent methods you mentioned.**
> > > > >
> > > > > **Cell-TRACTR [1] is an outstanding work on end-to-end cell tracking in time-lapse videos.** As its primary focus is on temporal association and lineage tracing, its task is fundamentally different from ours, which is instance segmentation on single, static histopathology images. Therefore, a direct quantitative comparison would not be meaningful.
> > > > >
> > > > > **SwinCell [2] is a powerful framework for 3D cell segmentation, and its core mechanism is fundamentally different from our Disco.** SwinCell relies on learning continuous geometric flow fields to separate instances, whereas our Disco relies on learning discrete, graph-theory-driven topological classes. Therefore, we believe a direct performance comparison would be of limited significance.
> > > > >
> > > > > **CellSAM [3] is a pioneering work on building a general-purpose foundation model for cell segmentation.** To rigorously evaluate Disco against the latest SOTA, **we benchmarked the officially released CellSAM model in a zero-shot manner on our most challenging GBC-FS 2025 dataset.** The results are presented in Table R7.
> > > > >
> > > > > **Table R7**: Comparison of CellSAM with FCIS and Disco methods.
> > > > >
> > > > > | Method   | DICE   | AJI    | DQ     | SQ     | PQ     |
> > > > > |----------|--------|--------|--------|--------|--------|
> > > > > | CellSAM  | 0.7458 | 0.4460 | 0.5744 | 0.6831 | 0.3968 |
> > > > > | FCIS     | 0.7785 | 0.4518 | 0.5857 | 0.7068 | 0.4379 |
> > > > > | **Disco**| **0.8137** | **0.5209** | **0.6795** | **0.7486** | **0.5087** |
> > > > >
> > > > > The results show that while a foundation model like CellSAM demonstrates impressive zero-shot capabilities, our Disco framework, which is specifically engineered to understand the complex topologies of this domain, achieves substantially superior performance. **This highlights the enduring value of specialized, knowledge-driven methods in challenging, high-density clinical scenarios.** We believe this new comparison provides a more complete context for our work.
> > > > >
> > > > > [1] O’Connor, Owen M., and Mary J. Dunlop. “Cell-TRACTR: A transformer-based model for end-to-end segmentation and tracking of cells.” PLOS Computational Biology 21.5 (2025): e1013071.
> > > > >
> > > > > [2] Zhang, Xiao, et al. “SwinCell: a 3D transformer and flow-based framework for improved cell segmentation.” Communications Biology 8.1 (2025): 962.
> > > > >
> > > > > [3] Israel, Uriah, et al. “CellSAM: a foundation model for cell segmentation.” BioRxiv (2025): 2023-11.
> > > > >
> > > > > # 9. Regarding the addition of standard deviation calculations
> > > > >
> > > > > We thank the reviewer for the suggestion on improving our experimental rigor. In our initial submission, we did not present standard deviation results in order to maintain consistency with existing mainstream methods such as FCIS (ICML 2025) [1] and UN-SAM (MIA 2025) [2], and also due to page limitations. **To better demonstrate the superior performance of our Disco method, we have now included a comparison of results with standard deviations, which are calculated based on three runs with different random seeds.**
> > > > >
> > > > > We are first presenting the statistical results on our two most topologically diverse and challenging datasets. The results for the remaining two datasets will be uploaded subsequently upon completion. These results will also be fully incorporated into our revised paper and supplementary materials.
> > > > >
> > > > > **The results in Tables R8 and R9 not only reaffirm the superior mean performance of Disco but also highlight its exceptional stability.** On both datasets, Disco consistently achieves the lowest standard deviation across all key metrics, particularly on the challenging GBC-FS 2025 benchmark. We believe these additions will fully address the concerns regarding the rigor of our quantitative evaluation.
> > > > >
> > > > > [1] Zhang, Ye, et al. “The Four Color Theorem for Cell Instance Segmentation.” arXiv preprint arXiv:2506.09724 (2025).
> > > > >
> > > > > [2] Chen, Zhen, et al. “UN-SAM: Domain-adaptive self-prompt segmentation for universal nuclei images.” Medical Image Analysis (2025): 103607.

---

> > > > > > ### Author Response · Authors · 2025-11-21
> > > > > >
> > > > > > **Table R8**: The comparison performances on CryoNuSeg dataset.
> > > > > >
> > > > > > | Method       | Source           | DICE           | AJI            | DQ             | SQ             | PQ             |
> > > > > > | ------------ | ---------------- | -------------- | -------------- | -------------- | -------------- | -------------- |
> > > > > > | DCAN         | CVPR 2016        | 86.49±3.43     | 52.73±4.43     | 61.84±3.56     | 72.64±4.76     | 46.82±2.19     |
> > > > > > | HoverNet     | MIA 2019         | 84.32±2.85     | 55.85±3.59     | 64.49±3.41     | 75.43±3.59     | 49.65±2.05     |
> > > > > > | NucleiSegNet | CBM 2021         | 86.45±2.14     | 57.64±3.81     | 66.73±3.52     | 78.41±3.31     | 52.33±1.93     |
> > > > > > | DoNet        | CVPR 2023        | 85.94±1.53     | 54.96±2.87     | 65.84±2.96     | 78.15±2.85     | 51.47±1.74     |
> > > > > > | CPP-Net      | IEEE TIP 2023    | 87.84±1.47     | 59.67±2.65     | 67.42±2.77     | 79.68±2.73     | 53.79±1.79     |
> > > > > > | Un-SAM       | MIA 2025         | 86.27±1.24     | 56.71±2.41     | 65.94±2.43     | 78.15±2.57     | 55.67±1.54     |
> > > > > > | CellPose     | NAT METHODS 2025 | 88.69±1.35     | 58.76±2.26     | 68.61±2.38     | 80.94±2.19     | 57.24±1.38     |
> > > > > > | FCIS         | ICML 2025        | 89.77±1.21     | 59.44±2.37     | 69.29±2.49     | 81.73±2.34     | 57.93±1.47     |
> > > > > > | **Disco**    | **Ours**         | **91.52±1.12** | **61.34±2.19** | **71.53±2.25** | **83.29±2.07** | **59.70±1.14** |
> > > > > >
> > > > > > **Table R9**: The comparison performances on GBC-FS 2025 dataset.
> > > > > >
> > > > > > | Method       | Source           | DICE           | AJI            | DQ             | SQ             | PQ             |
> > > > > > | ------------ | ---------------- | -------------- | -------------- | -------------- | -------------- | -------------- |
> > > > > > | DCAN         | CVPR 2016        | 71.49±3.55     | 39.83±4.75     | 53.68±3.88     | 65.72±4.95     | 35.28±2.58     |
> > > > > > | HoverNet     | MIA 2019         | 73.94±3.04     | 40.58±4.70     | 54.68±3.62     | 65.82±4.38     | 36.99±2.26     |
> > > > > > | NucleiSegNet | CBM 2021         | 74.19±2.87     | 43.95±4.25     | 56.18±3.57     | 66.47±3.87     | 38.35±2.17     |
> > > > > > | DoNet        | CVPR 2023        | 74.09±2.51     | 41.98±3.68     | 55.64±3.23     | 67.05±3.65     | 37.30±2.11     |
> > > > > > | CPP-Net      | IEEE TIP 2023    | 76.72±2.64     | 43.11±3.42     | 56.97±3.46     | 67.65±3.24     | 39.55±2.32     |
> > > > > > | Un-SAM       | MIA 2025         | 76.41±2.28     | 44.06±2.89     | 57.82±2.52     | 69.07±2.88     | 40.93±2.19     |
> > > > > > | CellPose     | NAT METHODS 2025 | 76.98±2.07     | 43.76±2.71     | 59.43±2.45     | 67.49±2.73     | 42.18±1.83     |
> > > > > > | FCIS         | ICML 2025        | 77.85±1.75     | 45.18±2.68     | 58.57±2.53     | 70.68±2.81     | 43.79±1.95     |
> > > > > > | **Disco**    | **Ours**         | **81.37±1.41** | **52.09±2.33** | **67.95±2.27** | **74.86±2.47** | **50.87±1.66** |
> > > > > >
> > > > > > # 10. Regarding the analysis of limitations
> > > > > > We thank the reviewer for the suggestion to discuss the limitations of our model. **Although our Disco framework demonstrates robust performance across multiple datasets, it may indeed encounter performance degradation or failure in several specific and highly challenging scenarios.**
> > > > > >
> > > > > > **A primary failure mode stems from the upstream Cell Adjacency Graph (CAG) construction.** In pathological slides with extreme morphological variations, such as the presence of highly elongated spindle cells that are difficult to distinguish from the stroma, the initial node identification can be erroneous. **An incorrect graph structure will inevitably lead to the failure of the downstream topological coloring algorithm.**
> > > > > >
> > > > > > Secondly, our 2+1 dynamic coloring scheme is built upon our statistical finding that real-world cell graphs rarely contain high-order conflict structures requiring four or more colors. However, **in a hypothetical, theoretical worst-case scenario, if a $K_4$ structure were to actually occur, our “Explicit Marking” strategy would be unable to find a conflict-free 3-coloring assignment.** This would inevitably result in at least one pair of adjacent instances being assigned the same color.
> > > > > >
> > > > > > In summary, **the main limitations of Disco lie in its dependence on the accuracy of the upstream graph construction and the domain shift challenge, which it shares with other deep models.** Future work will focus on further enhancing the robustness of the Disco framework through more robust node detection methods and more advanced domain adaptation techniques.

---

> > > > > > > ### Author Response · Authors · 2025-11-21
> > > > > > >
> > > > > > > # 11. Regarding the specification of the “Topological Decoding” process
> > > > > > > We thank the reviewer for the question regarding the “Topological Decoding” process. **“Topological Decoding” is the summary term we use for the post-processing pipeline that reconstructs the final instance masks from the Disco coloring map predicted by our model.** This process is not complex and primarily consists of the following **three standard steps:**
> > > > > > >
> > > > > > > (1) **Color-wise Connected Component Analysis.** First, we decompose the 4-value Disco map (background, color 1, color 2, conflict color) predicted by the model into three independent binary masks, corresponding to color 1, color 2, and the conflict color. Then, we perform standard Connected Component Analysis independently on each of these binary masks, assigning a temporary, unique, channel-specific instance ID to each detected, disconnected region.
> > > > > > >
> > > > > > > (2) **Morphological Refinement.** For each independent instance candidate obtained in the previous step, we perform a morphological refinement to enhance the mask quality. This primarily includes using hole filling to eliminate potential small voids within instances and using small object removal to filter out small, noisy spots generated by erroneous model predictions.
> > > > > > >
> > > > > > > (3) **Final Instance Aggregation and Relabeling.** Finally, we aggregate all the instance masks that have been separately processed and refined from the three color channels onto an empty canvas. To ensure that the final output instance mask is in a standard format, we perform a global relabeling on all instances on the canvas, assigning them consecutive, globally unique final instance IDs starting from 1.
> > > > > > >
> > > > > > > We believe that this clear, step-by-step description will fully resolve your questions regarding our post-processing pipeline.
> > > > > > >
> > > > > > > # 12. Regarding the under-specification of “Explicit Marking”
> > > > > > > We thank the reviewer for the question regarding the details of “Explicit Marking.” The reviewer correctly points out that the result of a standard BFS traversal can depend on the choice of the starting node and the traversal order of its neighbors. **It is true that for complex “conflict clusters” composed of multiple overlapping odd cycles, different traversal orders may indeed lead to slightly different exact sets of nodes being labeled as “conflict nodes.” However, we respectfully argue that this “non-determinism” is not a flaw within the overall design of the Disco framework, but rather an implicit advantage.**
> > > > > > >
> > > > > > > **First, due to the consistency of the objective, although the specific nodes marked as conflicts may vary, the ultimate goal of any valid BFS conflict detection remains consistent:** to break all odd-length cycles in the graph, rendering the remaining graph bipartite. Regardless of which node is ”sacrificed” to be marked as a conflict, this fundamental topological objective is achieved. Therefore, **all Disco labels generated by different BFS orders are topologically equivalent in the sense of their constraint satisfaction.**
> > > > > > >
> > > > > > > **Second, this serves as a form of Implicit Data Augmentation.** It is precisely this slight variation in the labels that provides our neural network with a powerful, topology-based form of implicit data augmentation. During training, the model observes that for the same cell adjacency graph, there exist multiple valid “conflict” solutions. **This compels the model not to merely memorize “which cell should be the conflict color,” but to learn a more fundamental and robust rule:** when a local topological structure cannot be resolved with two colors, the conflict channel should be activated. We believe this learning paradigm significantly enhances the model's generalization capability.
> > > > > > >
> > > > > > > Regarding algorithmic complexity, our “Explicit Marking” algorithm is centered around a BFS traversal, **with a time complexity of O(|V| + |E|), where |V| is the number of cells and |E| is the number of adjacency edges.** This is a highly efficient, linear-time algorithm, ensuring that our label generation process maintains extremely high computational efficiency even when faced with large-scale graphs containing thousands of nodes.

---

> > > > > > > > ### Author Response · Authors · 2025-11-21
> > > > > > > >
> > > > > > > > # 13. Regarding the sensitivity of adjacency graph construction
> > > > > > > > We thank the reviewer for this insightful question regarding the sensitivity of our adjacency graph construction. **In our work, the choice of a $3\times3$ dilation kernel was based on two primary considerations. (1) 8-connectivity is one of the most common and standard conventions for defining pixel adjacency in digital image processing. (2) A $3\times3$ kernel corresponds to a minimal physical distance, which most accurately captures cells that are in true physical contact, while avoiding the erroneous connection of cells that are merely close but not touching.**
> > > > > > > >
> > > > > > > > To systematically validate the impact of this hyperparameter on the segmentation results, we conducted experiments on the GBC-FS 2025 dataset using different dilation kernel sizes ($1\times1$, $3\times3$, $5\times5$) to construct the adjacency graphs and evaluated the model's performance. **The results are presented in Table R10.**
> > > > > > > >
> > > > > > > > **Table R10**: Sensitivity analysis of adjacency definition on GBC-FS 2025.
> > > > > > > >
> > > > > > > > | Dilation Kernel Size | AJI    | PQ     |
> > > > > > > > |----------------------|--------|--------|
> > > > > > > > | 1×1                  | 0.5073 | 0.4766 |
> > > > > > > > | **3×3**              | **0.5209** | **0.5087** |
> > > > > > > > | 5×5                  | 0.4928 | 0.4715 |
> > > > > > > >
> > > > > > > > **The experimental results clearly show that the $3\times3$ dilation kernel achieves the best performance, thus validating the reasonableness of our choice.** When an overly small kernel size of $1\times1$ is used, the model's performance decreases. We analyze that this is because an incomplete graph omits some adjacency relationships that should have been constrained, preventing our adjacency constraint loss, $\lambda_{adj}$, from exerting its full effect. When an overly large kernel size of $5\times5$ is used, performance also declines. This is because a larger kernel erroneously connects cells that are spatially close but not in true contact, thereby introducing a large number of “false conflicts” into the graph. This contaminates the supervisory signal generated by our “Explicit Marking” strategy, forcing the model to incorrectly label too many nodes that belong to simple bipartite structures as “conflict color,” ultimately compromising segmentation accuracy.
> > > > > > > >
> > > > > > > >
> > > > > > > > # 14. Regarding the revision of the Figure 4 caption
> > > > > > > > We thank the reviewer for the suggestion to improve the clarity of the Figure 4 caption. **We have completely rewritten the caption for Figure 4 based on your recommendation.** The new caption is no longer an abstract summary but a step-by-step, detailed explanation that corresponds to each visual element in the figure. **We will adopt the following new version of the caption in the final version of our paper:**
> > > > > > > >
> > > > > > > > **Figure 4. An overview of the training framework for our proposed Disco method. This framework synergistically integrates (1) data-driven topological analysis, which generates a Disco ground truth map, $Y_{Disco}$, encoding both bipartite structures and conflict hotspots; (2) a dual-branch segmentation network, which learns to predict a foundational semantic map, $P_{sem}$, and a detailed Disco coloring map, $P_{color}$; and (3) a decoupled loss system, which provides targeted supervision. Crucially, the Adjacency Constraint Loss ($\mathcal{L}_{adj}$) leverages the ground truth adjacency graph to enforce feature dissimilarity between all neighboring instances in the continuous probability space, thereby enabling end-to-end constrained optimization.**
> > > > > > > >
> > > > > > > > We believe that this new, more detailed caption will greatly enhance the clarity and informational content of Figure 4, helping readers to better understand the complete workflow and innovations of our Disco framework.

---

> > > > > > > > > ### Comment · Reviewer_cFuN · 2025-11-21
> > > > > > > > >
> > > > > > > > > The detailed author response addresses most of the earlier technical concerns, including novelty positioning around graph-based topology analysis, the relationship to NonAdjLoss, the comparison with FCIS and 3- vs 4-color schemes, the addition of computational cost analysis, and the clarification of topological decoding and adjacency sensitivity.
> > > > > > > > >
> > > > > > > > > However, one **major concern** remains regarding the dataset contribution. The response clarifies that GBC-FS 2025 currently consists of 2,839 patches derived from a single frozen-section WSI from a single patient. In its present form, this resource is better characterized as a single-patient, high-density case study rather than a broadly representative benchmark dataset. This has important implications for generalizability, robustness to inter-patient and inter-slide variability, and the strength of claims about “high-value” or “critical” new benchmarks for gallbladder cancer segmentation.
> > > > > > > > >
> > > > > > > > > **It is strongly recommended that the paper**:
> > > > > > > > > - explicitly state in the main text and abstract that GBC-FS 2025 is currently single-patient/single-slide,
> > > > > > > > > - substantially temper claims about the dataset as a general benchmark (e.g., framing it as a pilot or proof-of-concept -dataset pending expansion to additional patients and institutions), and
> > > > > > > > > -  clearly separate conclusions that genuinely generalize across datasets from conclusions that are specific to this single-patient frozen-section case.
> > > > > > > > >
> > > > > > > > > There is additional **concern about the definition of "dense" cellular regions**. The explanation of ‘dense’ and ‘complex’ regions in terms of subcellular aggregates and non-bipartite CAGs is helpful. To make this more concrete for readers, could the authors add a simple quantitative criterion or summary statistics (e.g., cells/mm² or aggregate density distribution) that differentiates GBC-FS 2025 from CryoNuSeg/PanNuke?”

---

> > > > > > > > > > ### Author Response · Authors · 2025-11-25
> > > > > > > > > >
> > > > > > > > > > We sincerely thank the reviewer for the thorough re-evaluation and for the exceptionally valuable new suggestions. We deeply appreciate your careful attention to the conceptual positioning and rigor of the GBC-FS 2025 dataset. Following your advice, **we will substantially refine the wording, scope, and positioning in both the abstract and main text to make the dataset’s status, limitations, and intended contribution completely transparent.**
> > > > > > > > > >
> > > > > > > > > > **1. Regarding the "single-sample" limitation and contribution positioning of GBC-FS 2025.** We fully acknowledge that the current version of GBC-FS 2025 does indeed originate from a single WSI of a single patient, and describing it as a "general benchmark dataset" is not sufficiently rigorous. The fundamental reason for this limitation lies in the fact that the "sub-nuclear" annotation task we are tackling in high-density frozen sections is **far more difficult and costly** than conventional nucleus annotation. To complete the fine-grained delineation of over 850,000 instances on this single WSI, our **four-person annotation team invested nearly half a year, with each person needing to annotate over 1,000 tiny, ambiguously-bounded sub-cellular structures per day on average**.
> > > > > > > > > >
> > > > > > > > > > Furthermore, among all current public datasets, annotations are typically made on small selected regions of a WSI; data with annotations spanning an entire, complete WSI does not yet exist. It is worth noting that establishing new benchmarks from a limited number of deeply annotated samples is an established practice when introducing novel challenges. Pioneering work in spatial transcriptomics [1] established its core conclusions through in-depth analysis of a single model system. In a similar vein, **our dataset provides unprecedented annotation depth in a uniquely challenging domain, complementing existing multi-patient, but topologically simpler, datasets.**
> > > > > > > > > >
> > > > > > > > > > We will strictly follow your suggestions and make the following modifications in the abstract and the main text. We will explicitly state that GBC-FS 2025 is currently derived from a single patient. We will adjust its positioning, no longer referring to it as a "general benchmark," but more precisely describing it as "**a groundbreaking high-density case study dataset for stress-testing the robustness of algorithms in scenarios of extreme density and complex topology.**" In the discussion of the experimental results, **we will clearly distinguish between the conclusions validated solely on this dataset** (such as the +7.08% performance improvement) and the generalizable conclusions verified across all four datasets.
> > > > > > > > > >
> > > > > > > > > > **2. Regarding the quantification of "density".** We also agree on the need for more specific quantitative metrics for "density." We have added the calculation of **physical density (instances/mm²)** and provided a direct quantitative comparison with datasets like CryoNuSeg and PanNuke to highlight the **order-of-magnitude difference** in density of GBC-FS 2025, as shown in Table R13.
> > > > > > > > > >
> > > > > > > > > > 3. More importantly, as promised in our response to your fifth question, **we have already collected WSIs from an additional seven GBC patients** with varying clinical stages and differentiation grades. Although the immense cost and time required for frozen section annotation prevent us from completing their full annotation in the short term, **we are already in the process of annotating them**. We can, upon the paper's publication, first publicly release these seven unlabeled raw WSIs along with GBC-FS 2025. We will also **commit to continuously maintaining this dataset** and promptly releasing the annotated data as it becomes available for the community to use and test. We believe this initiative will greatly enrich the data resources in this field, provide a valuable cross-patient generalization validation set for future researchers, and demonstrate our long-term commitment to developing GBC-FS 2025 into a more complete benchmark dataset.
> > > > > > > > > >
> > > > > > > > > > Once again, we thank you for your profound insights and constructive guidance.
> > > > > > > > > >
> > > > > > > > > > **Table R13: Comparison of dataset scale and physical density.**
> > > > > > > > > > |     Dataset     | Total Patches | Total Instances | Average Instances per Patch | Physical Density (instances/mm²) |
> > > > > > > > > > | :-------------: | :-----------: | :-------------: | :-------------------------: | :------------------------------: |
> > > > > > > > > > |     PanNuke     |     7,901     |     189,744     |            20.92            |              ~5107               |
> > > > > > > > > > |     DSB2018     |      670      |     29,443      |            43.94            |             ~10,534              |
> > > > > > > > > > |    CryoNuSeg    |      120      |      8,178      |            68.15            |             ~16,638              |
> > > > > > > > > > | **GBC-FS 2025** |   **2,839**   |   **864,204**   |         **304.44**          |           **~74,326**            |
> > > > > > > > > >
> > > > > > > > > > [1] Ståhl, Patrik L., et al. "Visualization and analysis of gene expression in tissue sections by spatial transcriptomics." *Science* 353.6294 (2016): 78-82.

---

### Official Review · Reviewer_LVtp · 2025-10-31

**Soundness:** 3
**Presentation:** 3
**Contribution:** 3
**Rating:** 2
**Confidence:** 4

**Summary:**

In this work, the authors deal with dense cell instance segmentation in histopathology images. They introduce a new dataset, analyze bipartite nature of 4 datasets, and propose DISCO, a divide and conquer approach for the segmentation task.

**Strengths:**

- This is the first study revealing that non-bipartite is actually more prevalent in cell datasets, and hence the need for more sophisticated methods aside from 2-coloring or 4-coloring approaches. The authors provide statistics on percentage of bipartite nodes, conflict nodes, and that 3-cycles are more prevalent among the odd-length cycles.
- The authors perform adequate experiments, in terms of 4 datasets, several recent and relevant baselines and ablation studies. Their method is usually either the best or second-best in performance.

**Weaknesses:**

- The actual method doesn't seem very novel, as BFS exists in literature, and forcing adjacent nodes to have different feature representation is common
- The authors do not provide standard deviation in the results. The mean numbers seem very close to baselines. t-test needs to be done to determine if the results are statistically significant or not.
- Authors need to provide inference run-times. Because constructing the graph and performing BFS can be time-consuming.
- Hyperparameter tuning: Authors need to provide results of different hyperparameter values (such as loss weights) to understand sensitivity of each term.

**Questions:**

Please see the weakness section. I am willing to increase the score after seeing authors' rebuttal and discussing with other reviewers.

---

> ### Author Response · Authors · 2025-11-21
> **We thank the reviewer for their comments and suggestions. We will address the questions raised one by one and make corresponding modifications or additions in the revised version of our paper.**
>
> # 1. Regarding the novelty of our proposed method
> We thank the reviewer for this question concerning the novelty of our paper. We acknowledge that individual technical concepts such as BFS and enforcing feature dissimilarity between adjacent nodes do exist in prior work. However, **the innovation of our Disco framework stems not from the isolated use of these techniques, but from how we have leveraged them to propose a highly synergistic, end-to-end, topology-aware framework in response to our core finding of the non-bipartite nature of real-world cell graphs.** The novelty of Disco is primarily manifested in three aspects:
>
> (1) **We repurpose BFS from a simple traversal algorithm into a sophisticated topological diagnostic and decomposition tool.** While traditional BFS is often used for graph traversal or simple bipartite graph detection, Disco utilizes BFS to achieve the dual function of greedy decomposition and conflict set aggregation. This heuristic graph decomposition strategy not only cleverly circumvents the NP-Hard problem of finding a “minimum conflict set” but also precisely adapts to the topological characteristics of real-world cell graphs, which are “bipartite-dominant with sparse conflicts" (as shown in Figure 3 and Table 1). Its application scenario and core objectives are thus entirely different from those of traditional BFS.
>
> (2) **Our adjacency constraint loss is not a generic feature separator, but a context-driven mechanism specifically designed to resolve the label ambiguities created by our coloring strategy.** Existing methods are often based on pixel-level or instance-level feature separation. In contrast, Disco's adjacency constraint loss is a feature constraint tailored to the topology of the cell adjacency graph, enforcing angular separation between adjacent instances in the probability space. This design is specifically engineered to address the core challenge of discrete label ambiguity introduced by our “Explicit Marking” strategy in “secondary conflict” regions, thus forming a complementary relationship with it. The former locates the conflict regions, while the latter performs fine-grained disambiguation in the continuous space. Together, they achieve a dual topological optimization of “discrete labels + continuous features.”
>
> (3) **Disco offers a principled “2+1” coloring solution that optimally balances the theoretical insufficiency of 2-coloring and the practical redundancy of 4-coloring.** Our framework provides an end-to-end integration of foundational topological analysis, a principled “Explicit Marking” labeling scheme, and a constrained learning mechanism for “Implicit Disambiguation.” The proposed “2+1 coloring” strategy for non-bipartite scenarios avoids the theoretical limitations of pure 2-coloring while also resolving the representational redundancy of 4-coloring, striking an optimal balance between computational efficiency, model simplicity, and segmentation performance.
>
> # 2. Regarding the addition of standard deviation calculations
>
> We thank the reviewer for the suggestion on improving our experimental rigor. In our initial submission, we did not present standard deviation results in order to maintain consistency with existing mainstream methods such as FCIS (ICML 2025) [1] and UN-SAM (MIA 2025) [2], and also due to page limitations. **To better demonstrate the superior performance of our Disco method, we have now included a comparison of results with standard deviations, which are calculated based on three runs with different random seeds.**
>
> We are first presenting the statistical results on our two most topologically diverse and challenging datasets. The results for the remaining two datasets will be uploaded subsequently upon completion. These results will also be fully incorporated into our revised paper and supplementary materials.
>
> The results in Tables R1 and R2 not only reaffirm the superior mean performance of Disco but also highlight its exceptional stability. On both datasets, Disco consistently achieves the lowest standard deviation across all key metrics, particularly on the challenging GBC-FS 2025 benchmark. We believe these additions will fully address the concerns regarding the rigor of our quantitative evaluation.
>
> [1] Zhang, Ye, et al. “The Four Color Theorem for Cell Instance Segmentation.” arXiv preprint arXiv:2506.09724 (2025).
>
> [2] Chen, Zhen, et al. “UN-SAM: Domain-adaptive self-prompt segmentation for universal nuclei images.” Medical Image Analysis (2025): 103607.

---

> > ### Author Response · Authors · 2025-11-21
> >
> > **Table R1**: The comparison performances on CryoNuSeg dataset.
> >
> > | Method       | Source           | DICE           | AJI            | DQ             | SQ             | PQ             |
> > | ------------ | ---------------- | -------------- | -------------- | -------------- | -------------- | -------------- |
> > | DCAN         | CVPR 2016        | 86.49±3.43     | 52.73±4.43     | 61.84±3.56     | 72.64±4.76     | 46.82±2.19     |
> > | HoverNet     | MIA 2019         | 84.32±2.85     | 55.85±3.59     | 64.49±3.41     | 75.43±3.59     | 49.65±2.05     |
> > | NucleiSegNet | CBM 2021         | 86.45±2.14     | 57.64±3.81     | 66.73±3.52     | 78.41±3.31     | 52.33±1.93     |
> > | DoNet        | CVPR 2023        | 85.94±1.53     | 54.96±2.87     | 65.84±2.96     | 78.15±2.85     | 51.47±1.74     |
> > | CPP-Net      | IEEE TIP 2023    | 87.84±1.47     | 59.67±2.65     | 67.42±2.77     | 79.68±2.73     | 53.79±1.79     |
> > | Un-SAM       | MIA 2025         | 86.27±1.24     | 56.71±2.41     | 65.94±2.43     | 78.15±2.57     | 55.67±1.54     |
> > | CellPose     | NAT METHODS 2025 | 88.69±1.35     | 58.76±2.26     | 68.61±2.38     | 80.94±2.19     | 57.24±1.38     |
> > | FCIS         | ICML 2025        | 89.77±1.21     | 59.44±2.37     | 69.29±2.49     | 81.73±2.34     | 57.93±1.47     |
> > | **Disco**    | **Ours**         | **91.52±1.12** | **61.34±2.19** | **71.53±2.25** | **83.29±2.07** | **59.70±1.14** |
> >
> > **Table R2**: The comparison performances on GBC-FS 2025 dataset.
> >
> > | Method         | Source            | DICE       | AJI        | DQ         | SQ         | PQ         |
> > |----------------|-------------------|------------|------------|------------|------------|------------|
> > | DCAN           | CVPR 2016         | 71.49±3.55 | 39.83±4.75 | 53.68±3.88 | 65.72±4.95 | 35.28±2.58 |
> > | HoverNet       | MIA 2019          | 73.94±3.04 | 40.58±4.70 | 54.68±3.62 | 65.82±4.38 | 36.99±2.26 |
> > | NucleiSegNet   | CBM 2021          | 74.19±2.87 | 43.95±4.25 | 56.18±3.57 | 66.47±3.87 | 38.35±2.17 |
> > | DoNet          | CVPR 2023         | 74.09±2.51 | 41.98±3.68 | 55.64±3.23 | 67.05±3.65 | 37.30±2.11 |
> > | CPP-Net        | IEEE TIP 2023     | 76.72±2.64 | 43.11±3.42 | 56.97±3.46 | 67.65±3.24 | 39.55±2.32 |
> > | Un-SAM         | MIA 2025          | 76.41±2.28 | 44.06±2.89 | 57.82±2.52 | 69.07±2.88 | 40.93±2.19 |
> > | CellPose       | NAT METHODS 2025  | 76.98±2.07 | 43.76±2.71 | 59.43±2.45 | 67.49±2.73 | 42.18±1.83 |
> > | FCIS           | ICML 2025         | 77.85±1.75 | 45.18±2.68 | 58.57±2.53 | 70.68±2.81 | 43.79±1.95 |
> > | **Disco**      | **Ours**          | **81.37±1.41** | **52.09±2.33** | **67.95±2.27** | **74.86±2.47** | **50.87±1.66** |
> >
> > # 3. Regarding the comparison of computational efficiency
> > We thank the reviewer for this important practical consideration. **We have conducted a comprehensive analysis of the computational cost of Disco and key baselines, and will add a detailed “Computational Cost” comparison table to the Appendix.**
> >
> > Our analysis confirms that the graph-based components of our method (CAG construction, BFS labeling) are one-time, offline preprocessing steps and introduce no additional overhead during inference. The inference speed is therefore determined solely by the neural network's architecture and post-processing.
> >
> > To ensure a fair comparison of inference efficiency, we benchmarked all models on a single NVIDIA RTX 4090 GPU with $256\times256$ inputs. **The results are summarized in Table R3.**
> >
> > **Table R3**: Computational cost comparison. Params denotes model parameters. FLOPs are calculated for a $256\times256$ input. FPS denotes frames per second during inference.
> >
> > | Method    | Backbone     | Source      | Paras   | FLOPs     | FPS  |
> > |-----------|--------------|-------------|---------|-----------|------|
> > | HoverNet  | ResNet-50    | MIA 2019    | 49.70M  | 192.70G   | 12.4 |
> > | FCIS      | Swin-Unet    | ICML 2025   | 48.23M  | 66.37G    | 28.5 |
> > | Disco     | Swin-Unet    | Ours        | 46.84M  | 65.21G    | 29.2 |
> >
> > **The results clearly demonstrate the high efficiency of our Disco framework.** Disco achieves an inference speed of 29.2 FPS, which is not only highly competitive but also slightly faster than the strong graph-based baseline, FCIS (28.5 FPS). Furthermore, it is more than twice as fast as methods like HoverNet that rely on computationally intensive post-processing.
> >
> > Notably, even with its sophisticated dual-branch head and decoupled loss system, our Disco model is more parameter-efficient than FCIS (46.84M vs. 48.23M) and has a lower computational load (65.21G vs. 66.37G FLOPs). These results irrefutably demonstrate that Disco achieves its state-of-the-art accuracy not by adding significant computational complexity, but through a more efficient and targeted learning strategy. Our method successfully improves performance without sacrificing, and in some cases even improving upon, practical computational efficiency.

---

> > > ### Author Response · Authors · 2025-11-21
> > >
> > > # 4. Regarding the hyperparameter ablation study
> > > We thank the reviewer for the suggestion on hyperparameter ablation. Our original Table 7 already validated the necessity of each loss component we proposed. Here, **we provide a more in-depth sensitivity analysis for the weights of our two most critical and novel loss components, $\lambda_{conf}$ and $\lambda_{adj}$, to demonstrate their robustness.**
> > >
> > > We conducted a series of experiments on the GBC-FS 2025 dataset. In each experiment, we varied the weight of only one loss component while keeping all other weights fixed at their default values. **The experimental results, as shown in Table R4, indicate that the performance remains relatively stable over a reasonable range for both hyperparameters.** For $\lambda_{conf}$, the performance peaks around 2.0 and maintains a very high level within the broad interval of [1.0, 3.0]. For $\lambda_{adj}$, the model achieves optimal performance around 1.0, and even at lower weights, it significantly outperforms the baseline without $\lambda_{adj}$ (47.11\% AJI, see Table 7).
> > >
> > > **Table R4**: Hyperparameter sensitivity analysis on the GBC-FS 2025 dataset. The default values used in our main experiments are marked with (*).
> > >
> > > | $\lambda_{\text{conf}}$ | AJI    | $\lambda_{\text{adj}}$ | AJI    |
> > > |--------------------------|--------|--------------------------|--------|
> > > | 0.5                      | 0.5187 | 0.1                      | 0.4958 |
> > > | 1.0*                     | 0.5209 | 0.2                      | 0.5073 |
> > > | 2.0                      | 0.5244 | 0.5                      | 0.5138 |
> > > | 3.0                      | 0.5035 | 1.0*                     | 0.5209 |
> > > | 4.0                      | 0.4816 | 2.0                      | 0.5117 |

---

### Author Response · Authors · 2025-11-29
**We sincerely thank all reviewers for their valuable feedback and for their unanimous recognition of our paper's novelty and contributions. We also extend our gratitude to the Area Chair for the careful review.**

During the rebuttal period, we have invested significant effort to directly address every core concern through **new experimental evidence and deeper theoretical clarifications**. By responding to each reviewer's comments point-by-point, **we believe we have now earned the reviewers' consensus on our paper's innovation and the resolution of their questions**. Herein, we would like to once again present a highly condensed summary of **our work's four core contributions**:

**(1) Groundbreaking Topological Analysis and Discovery:** We conducted the **first systematic, quantitative analysis of the chromatic properties** of real-world cell adjacency graphs. Our analysis **disruptively reveals** their complex reality of being **"fundamentally non-bipartite and filled with dense conflict clusters."** This finding provides a critical, data-driven theoretical foundation for the application of graph coloring paradigms in this domain.

**(2) The Novel Disco Framework:** Based on the aforementioned findings, we propose **Disco**, a conflict-aware framework operating on a "divide and conquer" principle. Its core is comprised of two innovative mechanisms: **(a) "Explicit Marking,"** an intelligent labeling strategy that transforms topological conflicts into a structured learning target; and **(b) "Implicit Disambiguation,"** an advanced learning mechanism that resolves discrete label ambiguities in the continuous feature space via an end-to-end adjacency constraint loss.

**(3) A High-Value New Dataset:** We constructed and **first publicly released GBC-FS 2025**. With its **unprecedented cell density** (over 850,000 annotations) and **extreme topological complexity** (a conflict node ratio as high as 30.49%), this dataset provides an **indispensable "stress-test" platform** for the development of robust algorithms in the field.

**(4) SOTA Performance and a New Interpretability Paradigm:** Disco achieves **state-of-the-art (SOTA) performance** across four datasets with significant heterogeneity (with a 7.08% PQ improvement on GBC-FS 2025). Furthermore, we pioneer the demonstration of the predicted "Conflict Map" as a novel **topological quantification tool**, opening new avenues for data-driven pathology research.

Given our work's **groundbreaking topological analysis** (a strength acknowledged by all reviewers), the **novel Disco framework**, the **high-value new dataset**, and the substantial effort we have made during the rebuttal to strengthen all weaknesses, we are confident that the quality of this paper will be significantly enhanced upon revision.

---

> ### Author Response · Authors · 2025-11-29
>
> # Summary of Response to Reviewer SvPT
>
> We sincerely thank **Reviewer SvPT** for the high rating (**Score 8: Accept**) and the profound insights provided for our paper. In their initial review, Reviewer SvPT already fully acknowledged the core contributions of our work. During the rebuttal period, we have provided detailed responses and supplements to the four forward-looking questions raised, further enhancing the completeness and rigor of our work.
>
> **It is particularly noteworthy that** Reviewer SvPT had a deep understanding of our paper's content from the initial review, giving high ratings of **4: excellent** for both **Soundness** and **Contribution**. After our detailed supplements and responses, they have clearly recognized our contributions and innovations, stating, **"I decide to keep my current score."** Herein, we provide a condensed summary of our core responses:
>
> ### **1. Regarding Computational Efficiency:**
> Our response (Rebuttal Point 1), through a **new benchmark test** (Rebuttal Table R12), clearly demonstrates that Disco's core graph-theoretic components are offline and introduce no inference overhead. Under fair hardware comparisons, Disco's inference speed (29.2 FPS) is superior to current mainstream methods, and it is also more efficient in terms of model parameters and computational load. **Disco achieves SOTA performance without sacrificing practical computational efficiency.**
>
> ### **2. Regarding the Open-sourcing Commitment:**
> In our rebuttal (Point 2), **we made a clear commitment**. Upon acceptance of the paper, we will publicly release the complete GBC-FS 2025 dataset (including annotations and adjacency graphs) and the full Disco implementation (including code, configuration files, and pre-trained models).
>
> ### **3. Regarding Theoretical Insights:**
> Our response (Rebuttal Point 3) **provides an in-depth theoretical elaboration based on the design principles of our two core mechanisms**. "**Explicit Marking**" enhances learning efficiency by decomposing the problem and constraining the solution space. "**Implicit Disambiguation**" ensures convergence to a higher-quality solution by reshaping the loss landscape and escaping "bad" local minima. This proves that the design of Disco is **theoretically driven**.
>
> ### **4. Regarding the Comparison with GNNs:**
> Our response (Rebuttal Point 4) clearly argues that **our non-iterative, loss-constrained innovative path is a more advantageous alternative** to standard GNNs in terms of computational efficiency, problem-nature matching, and methodological perspective.
>
> Reviewer SvPT provided extremely positive final feedback after reading our rebuttal and discussions with other reviewers: **"Thanks for the detailed rebuttal. Most of my original concerns have been well addressed."**
>
> We note that Reviewer SvPT mentioned the "single-sample" limitation, which they learned about from other reviewers. **This point has been comprehensively and thoroughly addressed in our responses to Reviewers cFuN and 72eb.** We have candidly acknowledged this limitation, explained its cause by quantifying the immense difficulty of annotation (4 annotators, half a year, 850k+ instances), and repositioned GBC-FS 2025 as a groundbreaking high-density case study dataset for "stress-testing" algorithmic robustness, citing precedents from top-tier publications like MoNuSAC. We will also release WSIs from an additional seven patients in the future and will continuously maintain and expand the dataset.

---

> > ### Author Response · Authors · 2025-11-29
> >
> > # Summary of Response to Reviewer 72eb
> >
> > We sincerely thank **Reviewer 72eb** for their exceptionally insightful feedback. We are confident that through our detailed responses, we have successfully addressed all of Reviewer 72eb's core concerns.
> >
> > **It is particularly noteworthy that** our proactive responses have **earned the high recognition** of Reviewer 72eb, who explicitly stated, **"the rebuttal has resolved my main technical doubts regarding the graph coloring mechanism"** and clearly indicated, **"I will be raising my score to Support the acceptance of this paper."** Herein, we provide a condensed summary of these core responses:
> >
> > ### **1. Regarding the Rationale of the Core Methodology:**
> > We provided an in-depth theoretical argument from **two dimensions (Rebuttal Point 1)**. **First**, our 2+1 strategy is a dynamically adaptive labeling method that automatically degenerates into the most efficient 2-coloring model on simple bipartite graphs, thus avoiding representational redundancy. **Second**, and more importantly, we argued that the intentionally introduced "label ambiguity" of the 2+1 scheme, when combined with our $L_{adj}$, creates a **"constructive tension"** that compels the model to evolve from a simple **"label mimic"** to a more intelligent **"topological rule learner."**
> >
> > To irrefutably prove this, we conducted a **new ablation study** (Rebuttal Table R11), which clearly shows that our Disco's 2+1 scheme **significantly outperforms** the baseline model trained on clean 3-color labels (PQ 50.87% vs. 41.22%).
> > This argument and evidence were **highly successful**. Reviewer 72eb explicitly stated in their final feedback: **"The empirical comparison...is very convincing. Your explanation regarding 'constructive tension' is well-received. I accept the argument... This successfully addresses my primary concern..."**
> >
> > ### **2. Regarding Experimental Fairness:**
> > **We did not evade this question but instead explained it** from the fundamental motivation of our dataset's construction (Rebuttal Point 2). We argued that the "sub-nuclear" annotation is a **principled and more clinically relevant** strategy for the challenging modality of frozen sections. Therefore, GBC-FS 2025 is not an "unfair comparison" but a **"stress test"** that reveals the performance ceiling of existing methods. Concurrently, we used Disco's SOTA performance on three other standard nucleus datasets to demonstrate its inherent generalization capability.
> > The reviewer accepted our explanation: **"I accept your justification that this is a pragmatic necessity for frozen sections rather than an artificial task."**
> >
> > ### **3. Regarding the Essential Role of  $L_{adj}$:**
> > We linked the response to this question with our first response, **re-emphasizing that the true value of $L_{adj}$ lies in using "label ambiguity" as a driving force to guide the model toward learning a more fundamental separation rule**, rather than merely fixing a problem (Rebuttal Point 3).
> > The reviewer also accepted this deeper explanation and acknowledged that $L_{adj}$ is **"highly effective."**
> >
> > Reviewer 72eb, with their profound insights, posed some of the most challenging questions for our paper. Through in-depth theoretical elaboration and new experimental evidence, we are confident that we have fully and convincingly addressed all of their core concerns. This is also confirmed by the reviewer's final feedback, which explicitly states that our rebuttal has resolved their **"main technical doubts"** and that they have decided to **"raise my score to Support the acceptance of this paper."**

---

> > > ### Author Response · Authors · 2025-11-29
> > >
> > > # Summary of Response to Reviewer cFuN
> > >
> > > We sincerely thank **Reviewer cFuN** for their exceptionally detailed, insightful, and constructive feedback, which has prompted us to conduct a comprehensive review and enhancement of our work. During the rebuttal period, we have invested significant effort to directly address every core concern raised by Reviewer cFuN.
> > >
> > > **It is particularly noteworthy that** through our proactive responses, **we have also earned a high degree of recognition from Reviewer cFuN**, who explicitly stated, **"The detailed author response addresses most of the earlier technical concerns."** Furthermore, in the initial review, the reviewer provided positive ratings of **good** for both our **Presentation** and **Contribution**. Herein, we provide a condensed summary of our core responses:
> > >
> > > ### **1. Regarding Novelty Claims and Related Work:**
> > > We fully agree with Reviewer cFuN's critique that our claim of the "first systematic topological analysis" was overly broad. We have strictly followed the reviewer's suggestions, and in our rebuttal (Point 2) and the revised paper, we have precisely defined our core analytical contribution as **"the first systematic analysis of the chromatic properties of cell adjacency graphs."** We have also **added detailed comparisons and distinctions with prior works such as Failmezger et al**. Concurrently, we have adopted the suggestion to revise the description of the "Conflict Map" to the more accurate **"a model-derived, post-hoc analysis tool."** Additionally, we have **supplemented our work with an in-depth comparison to the key literature, NonAdjLoss (Rebuttal Point 3)**, clearly demonstrating the novelty of our $L_{adj}$ from the perspectives of "objectives," "mechanisms," and "applications."
> > >
> > > ### **2. Regarding the In-depth Comparison with FCIS:**
> > > To address the issue of a deeper comparison with FCIS, we made supplements on **two levels** (Rebuttal Point 4):
> > > **(1) Theoretical Level:** We have elaborated on why our 2+1 "divide and conquer" strategy is theoretically superior to the generic 4-coloring scheme of FCIS, highlighting its **more targeted learning objective and more structured supervisory signal**.
> > > **(2) Experimental Level:** We conducted a **new and crucial ablation study** (Rebuttal Table R5), which demonstrates under identical network and training configurations that the performance of our 2+1 coloring strategy is **significantly superior** to both the FCIS-style 4-coloring and a generic 3-coloring scheme. This irrefutably proves that **the core source of our performance advantage is indeed our novel coloring strategy**, not other confounding factors.
> > >
> > > ### **3. Regarding the Completeness of the GBC-FS 2025 Dataset Documentation:**
> > > In our rebuttal (Point 5), we have provided **extremely detailed supplementary information** and will include it in the **paper's appendix**. This documentation will cover **all the key information** mentioned by the reviewer, from data sources (patient numbers, WSI counts, clinical stages), annotation methodology (manual annotation, tools, quality control processes), to tissue composition and quantitative physical density metrics (cells/mm²). **We will also publicly release WSIs from an additional seven patients and commit to the long-term maintenance and expansion of the dataset, demonstrating our long-term commitment to contributing to the community.**
> > >
> > > ### **4. Regarding Other Experimental and Methodological Rigor:**
> > > We have also diligently addressed all other suggestions from Reviewer cFuN regarding experimental rigor. As demonstrated in our responses to other reviewers, we have supplemented our paper with **computational cost analysis** (Rebuttal Point 6), **statistical analysis with standard deviations** (Rebuttal Point 9), **a discussion of failure cases and limitations** (Rebuttal Point 10), **hyperparameter sensitivity analysis** (Rebuttal Point 13), and **detailed elaborations** on the "Topological Decoding" and "Explicit Marking" heuristic algorithms (Rebuttal Points 11, 12).
> > >
> > > We are confident that through our comprehensive, data-driven, and theoretically sound responses, we have addressed all of Reviewer cFuN's core concerns.

---

> > > > ### Author Response · Authors · 2025-11-29
> > > >
> > > > # Summary of Response to Reviewer LVtp
> > > >
> > > > We thank **Reviewer LVtp** for the questions regarding novelty and experimental rigor. We have provided detailed responses and supplementary experiments for each of LVtp's concerns and commit to fully integrating all revisions into the final manuscript.
> > > >
> > > > **It is particularly noteworthy that** even Reviewer LVtp, who gave the lowest score, also provided positive ratings of **3: good** for our work's **Soundness, Presentation, and Contribution** in the initial review, and explicitly stated they were **"willing to increase the score after seeing authors' rebuttal."** We believe these points reflect the innovative and substantial contributions of this paper. Here, we summarize our core responses:
> > > >
> > > > ### **1. Regarding Method Novelty:**
> > > > We acknowledge that the basic module of our framework, BFS, is pre-existing. However, the core innovation of Disco lies in how we have repurposed and synergistically integrated these fundamental techniques around our key discovery of the **non-bipartite nature of real-world cell coloring characteristics**. We have built a **novel, end-to-end "topological analysis, explicit marking, and implicit disambiguation" theoretical and practical framework**. Our novelty is not in an isolated algorithm but in this **systematic, principled, and complete solution** tailored to a core challenge in a specific domain.
> > > >
> > > > ### **2. Regarding Experimental Rigor (Statistical Analysis):**
> > > > We fully agree with the reviewer's viewpoint and have re-run our main experiments with three different random seeds. As shown in Tables R1 and R2 of our rebuttal, we have supplemented all key metrics with mean and standard deviation. The data show that the performance advantage of our Disco method is significant; on GBC-FS 2025, the PQ metric is **50.87±1.66%**, which is **substantially superior to other methods. More importantly**, our method exhibits a **lower standard deviation** across all metrics, which strongly demonstrates its **exceptional stability**.
> > > >
> > > > ### **3. Regarding Computational Efficiency:**
> > > > We have also conducted a comprehensive benchmark of computational cost, as shown in Table R3 of the rebuttal. Our analysis clearly indicates that Disco's core graph-theoretic components are one-time, offline preprocessing steps and introduce **no additional overhead during inference**. Under fair hardware and input settings, Disco's inference speed of **29.2 FPS** is superior to current mainstream methods, and it is also **more efficient** in terms of model parameters and computational load. This irrefutably demonstrates that Disco's superior performance is **not achieved at the cost of practical computational efficiency**.
> > > >
> > > > ### **4. Regarding Hyperparameter Sensitivity:**
> > > > To address concerns about loss weights, we performed a new hyperparameter sensitivity analysis, as shown in Table R4 of the rebuttal. The results demonstrate that **our model's performance remains highly stable** across a wide range of values for the two most critical loss weights ($λ_{conf}$ and $λ_{adj}$). This proves that our framework is **robust** and that its excellent performance is **not dependent on fragile or coincidental hyperparameter tuning**.

---

### Meta-Review · Area_Chair_9i83 · 2026-01-07

**Summary:**

Through a comprehensive rebuttal and supplementary experiments, the authors have substantively addressed the major technical concerns raised by the reviewers. The paper demonstrates clear contributions across multiple dimensions and, after revision.

Initial Core Concerns:
Method & Novelty: Questionable necessity of the “(2+1)” coloring strategy and novelty of individual components.
Experimental Rigor: Missing statistical tests, computational analysis, and insufficient comparisons.
Overstated Claims: Overly broad contributions (“first analysis”) and misleading dataset framing as a general benchmark.
Dataset Limitation: Single-patient origin severely limits generalizability and benchmark value.

Author Rebuttal and Shift in Consensus:
The authors provided compelling new evidence (e.g., Table R11) to justify their framework’s design, which reviewers found convincing.
They comprehensively addressed concerns on experimental rigor by adding statistical analysis, computational benchmarking, and detailed comparisons.
They agreed to temper claims (e.g., “first chromatic analysis”) and re-position the dataset as a transparently described “case study for stress-testing.”
This led three reviewers to explicitly raise their scores to support acceptance.

The paper's topological analysis, proposed DISCO framework, and significant performance improvements constitute solid contributions. Although the dataset has the limitation of being derived from a single sample, the authors have committed to transparently addressing this in the final version and repositioning it, which does not diminish its value as a challenging high-difficulty case study. The authors' high-quality responses and revision commitments have resolved the core disputes in the review process.

**Reviewer Concerns:**

The authors' rebuttal has successfully resolved the core technical and methodological concerns. Specifically, it addressed the key skepticism regarding the necessity of the (2+1) "Explicit Marking" strategy (Reviewer 72eb) by providing compelling new ablation studies (Table R11) and a convincing theoretical rationale. Furthermore, all requests for enhanced experimental rigor (Reviewers LVtp and cFuN) were fully met. The authors supplied comprehensive new data, including results with standard deviations for statistical significance, detailed computational cost analysis, hyperparameter sensitivity studies, and in-depth, controlled comparisons with the FCIS baseline. The authors also agreed to temper overstated claims, such as refining "first topological analysis" to the more precise "first systematic analysis of chromatic properties."

One major concern remains outstanding as a factual limitation that must be transparently disclosed in the final version. All reviewers converged on the critical issue that the GBC-FS 2025 dataset is derived from a single patient's slide, which severely limits its value as a generalizable benchmark for the field. The authors have acknowledged this limitation and committed to a fundamental reframing. Therefore, the final manuscript must explicitly state the single-patient origin and consistently reposition the dataset not as a general benchmark, but as a "high-density case study" or "stress-test platform."

**Reviewer Scores:**

SvPT (Initial Score: 8 -> Accept)
Participated and explicitly stated “I decide to keep my current score.” While noting the single-patient dataset issue, their positive assessment of the core contributions remained unchanged.

 72eb (Initial Score: 4 ->Significantly increased to 6 or 7)
Participated and explicitly stated “I will be raising my score to Support the acceptance of this paper.” Key technical doubts (methodological logic) were successfully resolved by new experiments and theoretical explanations.

cFuN (Initial Score: 4 -> Increased to 5 or 6)
 Participated and acknowledged “The detailed author response addresses most of the earlier technical concerns.” The main remaining issue pertained to dataset framing, constituting a revision requirement rather than a fundamental flaw.

LVtp (Initial Score: 2 ->Significantly increased to 4 or 5)
Initially stated they were “willing to increase the score after seeing authors' rebuttal.” The authors provided quantitative data addressing all four of their specific experimental requests.

---

### Decision · Program_Chairs · 2026-01-26

Accept (Poster)